# Structure of the human systemic RNAi defective transmembrane protein 1 (hSIDT1) reveals the conformational flexibility of its lipid binding domain

Vikas Navratna[1,2], Arvind Kumar[3], Jaimin K Rana[1,2], Shyamal Mosalaganti[1,2,4]

In *Caenorhabditis elegans*, inter-cellular transport of the small non-coding RNA causing systemic RNAi is mediated by the transmembrane protein SID1, encoded by the *sid1* gene in the systemic RNAi defective (*sid*) loci. SID1 shares structural and sequence similarity with cholesterol uptake protein 1 (CHUP1) and is classified as a member of the ChUP family. Although systemic RNAi is not an evolutionarily conserved process, the *sid* gene products are found across the animal kingdom, suggesting the existence of other novel gene regulatory mechanisms mediated by small non-coding RNAs. Human homologs of *sid* gene products—hSIDT1 and hSIDT2—mediate contact-dependent lipophilic small non-coding dsRNA transport. Here, we report the structure of recombinant human SIDT1. We find that the extra-cytosolic domain of hSIDT1 adopts a double jelly roll fold, and the transmembrane domain exists as two modules—a flexible lipid binding domain and a rigid transmembrane domain core. Our structural analyses provide insights into the inherent conformational dynamics within the lipid binding domain in ChUP family members.

## Introduction

RNAi is post-transcriptional regulation of protein expression, wherein small non-coding double-stranded (ds) RNAs such as exogenous siRNA and endogenous miRNA target and induce degradation of homologous intracellular mRNA (1, 2, 3). Although this RNA homology–mediated gene silencing phenomenon is being explored to experimentally alter gene expression with academic and therapeutic applications, the mechanism of dsRNA transport has been poorly understood (4, 5, 6, 7). Most small non-coding dsRNA transport occurs in bulk by plasmodesmata in plants or clathrin-mediated endocytosis in other eukaryotes. Alternatively, specific small non-coding RNA transport processes mediated by

high-density lipoprotein binding scavenger receptors or lipophilic transporters have been reported (8, 9, 10). In *Caenorhabditis elegans*, systemic RNAi is mediated by the products of the genes in systemic RNAi defective (*sid*) loci. SID1 (systemic RNAi defective protein 1) is a transmembrane protein involved in the inter-cellular transport of small non-coding RNA that causes systemic RNAi (11, 12). In humans, there are two known systemic RNAi defective transmembrane proteins (SIDTs)—hSIDT1 and hSIDT2—that are implicated in hepatocellular lipid homeostasis, glucose tolerance, insulin secretion, tumor development, cancer chemoresistance, RNA translocation and degradation during autophagy, and activation of innate immunity (13, 14, 15, 16, 17, 18, 19).

hSIDT1 localizes on the plasma membrane, endolysosomes, and endoplasmic reticulum (13, 17, 20, 21, 22). The in vitro overexpression of hSIDT1 results in increased siRNA transport and subsequently siRNA-mediated gene silencing (23). Oncogenic miRNA, miR-21, induces resistance in tumors to gemcitabine, a nucleoside analog and inhibitor of DNA synthesis that is popularly used as a cancer therapeutic (20). hSIDT1 is believed to transport miR-21 into the tumor cell, promoting chemoresistance to gemcitabine. hSIDT1 lining the gastric pit cells in the stomach is indicated to uptake exogenous dietary miRNA in a pH-dependent manner, thereby enabling the internalized miRNA to regulate host gene expression (13). The homolog of hSIDT1, hSIDT2, transports siRNA into late endosomes and lysosomes. SIDT2 overexpression enhances siRNA transport, whereas knockdown reduces RNA degradation, and a knockout disturbs glucose homeostasis (15, 24). Like SIDT1, SIDT2 is also implicated in crucial biological processes such as lysosomal membrane permeability, hepatic lipid homeostasis, apoptosis, and tumor proliferation (14, 20, 25, 26). Despite SIDTs being a potential target for therapeutics against tumor progression, liver diseases, and type II diabetes, their exploration as potential drug targets has been impeded by the lack of high-resolution 3D structures of SIDTs.

The transporter classification database categorizes SIDTs as $\alpha$-type channels (TCDB: 1.A.79), belonging to the cholesterol uptake (ChUP) or dsRNA uptake family of transmembrane proteins whose function and stability are regulated by cholesterol (27). In fact, ChUP

[1]Life Sciences Institute, University of Michigan, Ann Arbor, MI, USA   [2]Department of Cell and Developmental Biology, University of Michigan, Ann Arbor, MI, USA   [3]Thermo Fisher Scientific, Waltham, MA, USA   [4]Department of Biophysics, College of Literature, Science and the Arts, University of Michigan, Ann Arbor, MI, USA

Correspondence: navratna@umich.edu; mosalaga@umich.edu

family members possess conserved transmembrane cholesterol recognition/interaction amino acid consensus motif (CRAC), suggesting a role in cholesterol binding and transport, and mutations in the CRAC motifs affect the localization of SIDTs (28, 29, 30). SIDTs are believed to operate as oligomers—whereas the full-length SIDTs have been shown to form dimers both in vitro and in cellulo; the N-terminal extra-cytosolic domain (ECD) of SIDTs has been shown to assemble as tetramers (31, 32, 33, 34, 35, 36, 37). ECDs of the SIDTs are believed to be crucial for substrate recognition and nucleic acid binding (31, 32, 38, 39). Qian et al. reported the structure of human SIDT2, where they show that SIDT2 exists as a dimer with a β-strand–rich ECD, and a transmembrane domain (TMD) comprised of 11 transmembrane helices (TMs). The dimer interface is formed predominantly by ECDs and partially by three TMs. They also suggest that the TM region of SIDT2 is capable of zinc-dependent lipase activity (33). Here, we use single-particle cryo-electron microscopy (cryo-EM) to determine the structure of heterologously expressed full-length hSIDT1 at ~3.4 Å. We observe that hSIDT1 exists as a dimer in vitro, where the dimer interface is carved by both the ECD and TMD. The β-strand–rich ECD adopts a double jelly roll conformation, and two such double jelly rolls, one from each protomer, are juxtaposed in a C2 symmetry to form a dimer. The TMD can be divided into a dynamic lipid binding domain (LBD) and a TMD core that is stabilized by four tiers of interactions. We also identify a phenylalanine highway within the TMD core, which is often seen in ATP-binding cassette (ABC) sterol transporters (40). Finally, by comparing our structure with the existing structures of the ChUP family of transporters, we highlight the intra-chain and inter-chain dynamics within the ChUP family of membrane proteins.

# Results

## Expression, purification of dimeric hSIDT1, and structure determination by cryo-EM

Purification and structural characterization of full-length human SIDTs have remained unexplored for a long time. We expressed full-length hSIDT1 fused to GFP on the C-terminus in HEK293S GnTI- cells by baculoviral transduction, along with a C-terminal Strep-II tag for purification (Table S1). To identify the detergent that is most suitable for purification of hSIDT1-GFP, we solubilized transiently transfected mammalian cells expressing hSIDT1-GFP in 1% detergent and analyzed the solubilized lysates by fluorescence-detection size-exclusion chromatography (FSEC; Fig S1A–E). We noticed that hSIDT1-GFP solubilized in digitonin and DMNG resulted in homogeneous samples. For cryo-EM sample preparation, we used baculoviral transduction to express large quantities of full-length hSIDT1-GFP. As ChUP family members contain CRAC motifs and are believed to be a cholesterol transporter, we excluded cholesterol and its analogs from our purification buffer to obtain the structure of an apo-hSIDT1. We chose digitonin to solubilize the large-scale mammalian cell membranes expressing hSIDT1-GFP and performed all the subsequent purification steps in digitonin to produce homogeneous dimeric hSIDT1-GFP fusion (Fig S1F–H). Henceforth, hSIDT1-GFP discussed in this report will be referred to

as hSIDT1. We used cryo-EM to determine the structure of hSIDT1 to a global resolution of ~3.4 Å, with the N-terminal ECD at a better local resolution than the C-terminal TMD (Table 1, Figs S2 and S3A–C). We were able to model the entire protein except for the first 40 amino acids in the disordered N-terminus, the cytosolic loops CL1 and CL4, a portion of β7′-β8′ hairpin (Q278-N286), and the GFP on the C-terminus (Fig S3D). Among the regions that were modeled, the resolution for TMs6–9 was the poorest, so we used the unsharpened map to place the Cα traces of these helices into the density and truncated the side chains of amino acids in these regions (Fig S3D, inset).

## Architecture of hSIDT1

hSIDT1 is a dimer where the protomers are related to each other by a twofold (C2) rotational symmetry, with the axis of rotation being perpendicular to the plane of the lipid bilayer (Fig 1A–C). Each protomer has an N-terminal β-sheet–rich ECD (yellow box) and a C-terminal all α-helical TMD, which can be divided into two parts—the TMD core (blue box) and the LBD (gray box) (Figs 1 and 2). All three regions participate in the assembly of the dimer (Fig 3). We notice that the architecture of hSIDT1 and hSIDT2 is similar, and the intra-chain and inter-chain interactions that stabilize the protein are conserved between the two homologs (Figs S4, S5, and S6).

### ECD

The hSIDT1 ECD contains a 260–amino acid domain (A42-I301) comprised of two tandem β-sandwiches arranged like a double jelly roll (Figs 2A and C and 3A). A classic jelly roll fold is an eight-stranded antiparallel β-sheet sandwich with the strands β1, β3, β6, and β8 forming the top sheet, such that the strand order of the top sheet is β1-β8-β3-β6 and of the bottom sheet is β2-β7-β4-β5 (39). Within the ECD of each SIDT1 protomer, we see two such jelly rolls, JR1 (G45 to K167) and JR2 (R172 to I301), connected in tandem by a flexible linker (H168 to L171) such that the top sheet of JR1 and the bottom sheet of JR2 are in the same plane (Figs 2 and 3A). The bottom strands of the double jelly roll line the SIDT1 dimer interface of the protein. JR1 and JR2 arrangement allows the formation of an extensive network of hydrogen bonds and aromatic ring stacking interactions between β6, β5-β6 hairpin, and β6-β7 hairpin of JR1, and β2′, β7′, β7′-β8′ hairpin, and β4′-β5′ hairpin of JR2. JR1 and JR2 are held together, toward the dimer interface, by a disulfide bond (C130–C222) between cysteines in β6-β7 hairpin and β4′-β5′ hairpin (Figs 3A and S4A). The JR1-JR2 interface is also supported by a network of hydrogen bonds formed between conserved residues, mainly R118, Y120, E124, S184, and E267 (Figs 3A, S5A, and S6A). The ECD is also stabilized by three pairs of salt bridges (R98-D148, H178-E294, and R289-D268) and has five predicted N-glycosylation sites—N57, N67, N83, N136, and N282. We used HEK293S GnTI cells that lack N-acetylglucosaminyltransferase I activity for recombinant hSIDT1 production, thus yielding a protein that lacks complex N-glycosylation. Although we observe density for the N-acetyl-glucosamine at the Asn residues that are predicted glycosylation sites, our resolution is relatively poor to model N-acetylglucosamine residues in the density unambiguously.

**Table 1. hSIDT1 cryo-EM data collection, processing, and validation statistics.**

| | EMD-42943 and PDB-8V38 |
|---|---|
| **Data collection and processing** | |
| Magnification | 105,000x |
| Voltage (kV) | 300 |
| Data collection mode | Super-resolution |
| Electron exposure (e−/Å$^2$) | 50 |
| Defocus range (μm) | −1.2 to −2.5 |
| Physical pixel size (Å) | 0.848 |
| Symmetry imposed | C2 |
| Initial particle images (no.) | 3,204,173 |
| Final particle images (no.) | 122,683 |
| Map resolution (unmasked, Å) at FSC$_{0.143}$ | 3.98 |
| Map resolution (masked, Å) at FSC$_{0.143}$ | 3.44 |
| Map resolution range (local resolution) | 2.0–8.0 |
| **Refinement** | |
| Map sharpening B factor (Å$^2$) | −30 |
| **Model composition** | |
| Chains | 2 |
| Atoms | 9,176 (hydrogens: 0) |
| Residues | Protein: 1,236 nucleotide: 0 |
| Water | 0 |
| Ligands | 0 |
| **Bonds (RMSD)** | |
| Length (Å) (# > 4; Sigma-Aldrich) | 0.004 (0) |
| Angles (°) (# > 4; Sigma-Aldrich) | 0.907 (4) |
| **MolProbity score** | 1.61 |
| Clash score | 2.61 |
| **Ramachandran plot (%)** | |
| Outliers | 0.00 |
| Allowed | 9.24 |
| Favored | 90.76 |
| **Rama-Z (Ramachandran plot Z-score, RMSD)** | |
| whole (N = 1254) | −2.79 (0.22) |
| helix (N = 482) | −1.22 (0.22) |
| sheet (N = 254) | 0.34 (0.33) |
| loop (N = 518) | −3.65 (0.22) |
| Rotamer outliers (%) | 1.13 |
| Cβ outliers (%) | 0.00 |
| **Peptide plane (%)** | |
| Cis proline/general | 4.5/0.0 |
| Twisted proline/general | 4.5/0.0 |
| Cα BLAM outliers (%) | 6.57 |
| **ADP (B-factors)** | |
| Iso/Aniso (#) | 9176/0 |

**Table 1.  Continued**

|  | EMD-42943 and PDB-8V38 |
| --- | --- |
| Protein (min/max/mean) | 47.35/263.37/116.76 |
| Model versus data |  |
| CC (mask) | 0.75 |
| CC (box) | 0.52 |
| CC (peaks) | 0.49 |
| CC (volume) | 0.72 |

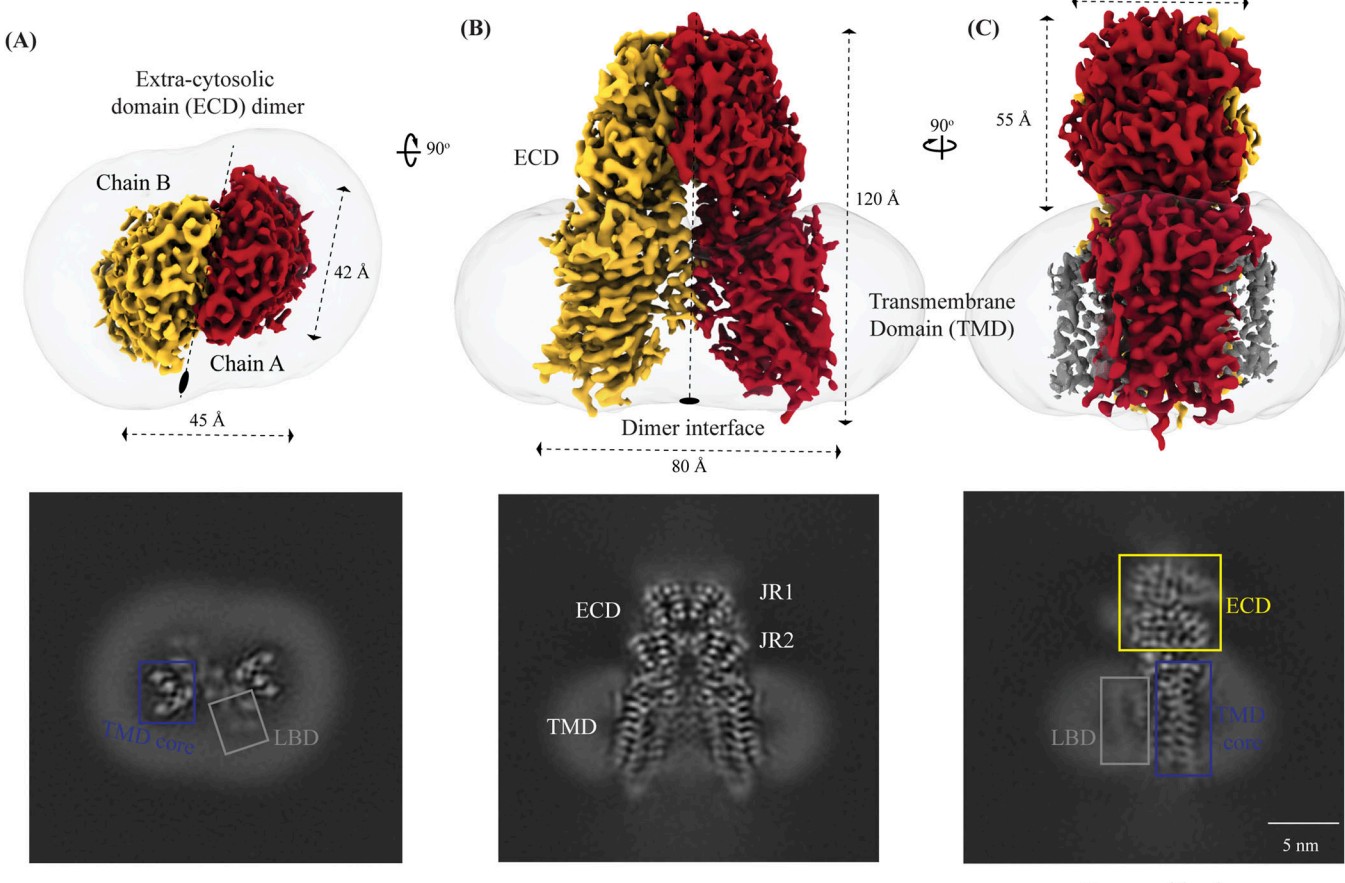

**Figure 1.   Cryo-EM structure of the hSIDT1 dimer.**
**(A, B, C)** show the extra-cytosolic domain view, broad side view, and narrow side view, respectively, of the ~3.4 Å cryo-EM map of the hSIDT1 dimer. Chains A and B are displayed in red and yellow, respectively, at an isosurface threshold level of 0.14 in ChimeraX (44). The non-contiguous density of lipid binding domain (LBD) is highlighted as a gray mesh in panel **(C)**. Micelle is shown in transparent gray. The bottom half of panels A-C shows cross-sections of the final 3D map from CryoSPARC, in three views, displayed using IMOD, indicating the poor resolution of the lipid binding domain (gray box) compared with the transmembrane domain core (blue box) or ECD (41, 48).

## TMD

The TMD of the SIDT1 protomer is about 500 amino acids long and comprises 11 TM helices. TMs1–4, along with TM10 and TM11, form the TMD core of the protein, and TMs5–9 form the LBD (Fig 2). TM1 and TM2 are connected by a disordered cytoplasmic loop (CL1) that is over 100 amino acids and is believed to form a cytosolic domain (CD) with an RNA binding ability (Fig 2A) (31, 32, 38). Owing to its flexibility, we do not see density for CL1 in our structure. The ECD sits

atop the TMD core via JR2-TMD core interactions and does not interact with the LBD (Fig 3B).

**TMD core**  The TMD core is stabilized by four tiers of evolutionarily conserved interactions—the TMD-ECD interface, the TMD disulfides, a metal ion binding site, and a phenylalanine highway (Figs 3B and C and 4B). An extensive network of conserved hydrogen bonds, electrostatic interactions, and salt bridges mediated by K203, Q238,

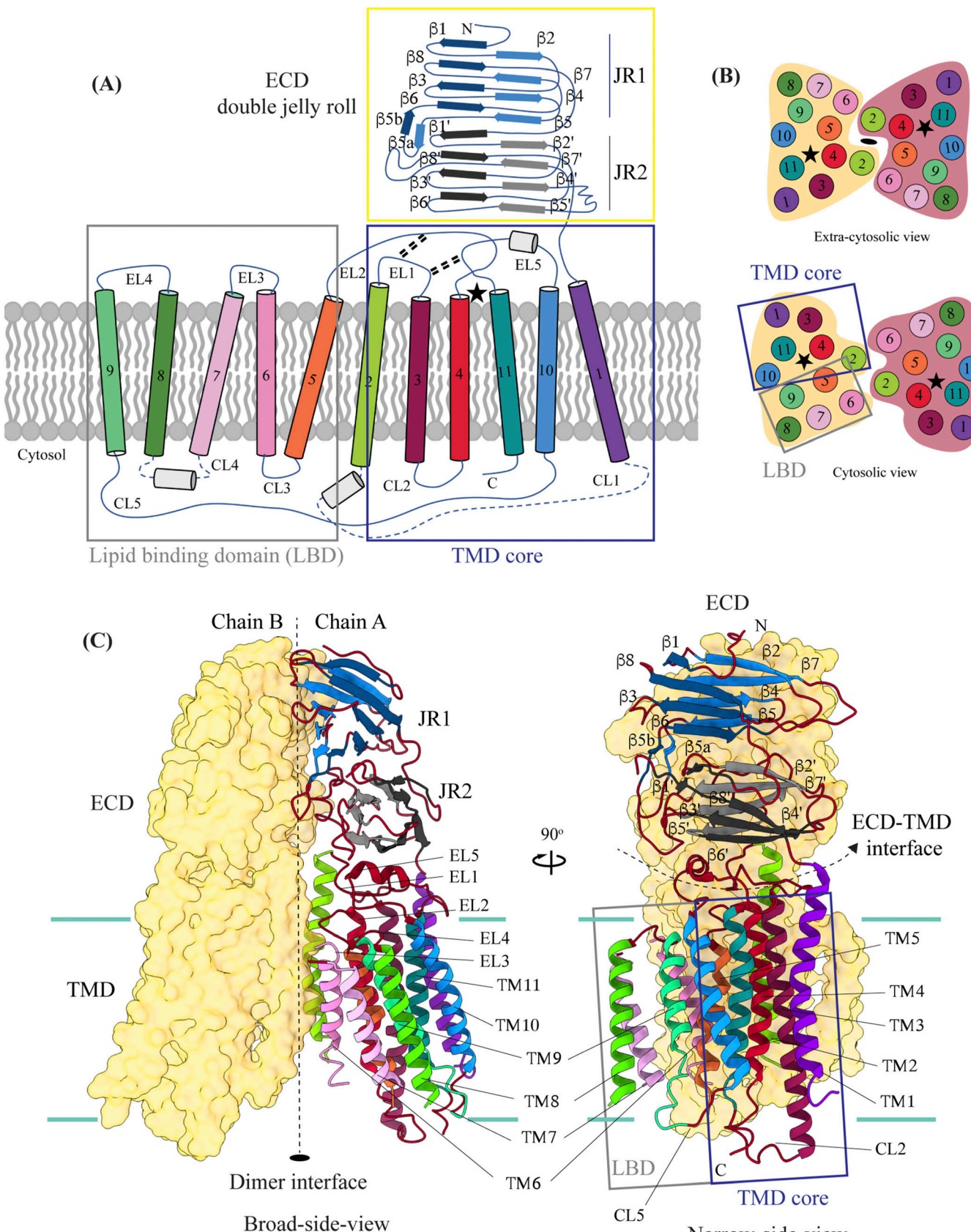

▶▶▶▶▶ Life Science Alliance

S239, T241, C271, Q476, D477, R776, E777, and E781 at the TMD-ECD interface between EL1, EL5, β3'–β4' hairpin, β5', β6', and β7'–β8' hairpin position the ECD over the TMD core such that the access to the metal binding site from the extra-cytosolic space is blocked by the ECD (Figs 2A and B, 3B, S4B, S5B, and S6B). At the TMD-ECD interface, the extra-cytosolic side of the TMD core—TMs2–4, TM10, and TM11—is held together by two disulfides (C479–C565 and C485–C782) formed between EL1, EL2, and EL5 (Figs 2A, 3B, and 4B). The TMs3–5 and TM11 carve a conserved metal ion binding site lined by amino acids N501, S559, H563, D574, H791, and H795 (Figs 3C, S5C, and S6C). This site has been suggested to be a zinc site and is believed to be essential for the activity of SIDT-like proteins (33, 34, 35, 36, 37). Although we observe density coordinated by H563, D574, H791, and H795 in our structure, because of a lack of convincing biochemical evidence, we have not modeled a metal ion at this site in the structure (Fig 3C). In hSIDT1, we notice a pouch, facing the lipid bilayer, formed by TM1, TM3, TM10, and TM11. We find a series of ordered lipids and detergents bound to this region in our cryo-EM density map (Fig 4A). However, because of the relatively poor local resolution in this area, we have not modeled any lipid or detergent in this density (Fig S3C). ABC sterol transporters such as the ABCG family have a conserved structural motif called a phenylalanine highway, where a series of phenylalanines line the dimer interface and enable binding of cholesterol. Like ABCG, these phenylalanine highways have been suggested to exist in other cholesterol binding proteins such as NPC1, PTCH1, and ChUP family of proteins, albeit not always in the dimer interface (40). In SIDT1, we notice a similar phenylalanine highway within the pouch formed by TM1, TM3, TM10, and TM11 (Fig 4B). Especially, TM11 (D786-L810) and TM1 (K302-R334) are studded with six and five phenylalanine residues, respectively. The aromatic ring stacking interactions in this pouch form the fourth tier of interactions that stabilize the TM core along with the disulfides, metal ion binding site, and the TMD-ECD interface interactions.

**LBD** TMs5–9 form a conformationally flexible domain, which is poorly resolved in our cryo-EM map (Figs 1C and 4A and S3). Henceforth, we will refer to this region of the protein as the LBD (Figs 1 and 2). The local resolution for the LBD in our structure is poorest of all regions, especially for TMs6–8 (Figs 1 and S3C and D). As a result, we have modeled only the main chain Cα for most part of the helices in this region. TM5 and TM6 of the LBD line the dimer interface along with TM2 of the TMD core. TM7 and TM8 lie on the periphery of the protein and protrude into the lipid bilayer away from the TMD core. TM5 and TM9 align with the TMD core, and at the interface of this LBD and TMD core, we also notice unexplained ordered density, which could be either lipid or detergent (Figs 1 and 4A).

### hSIDT1 dimer interface

SIDT1 dimer interface spans ~2,800 Å$^2$, and it can be divided into three sub-interfaces (SI)—1, 2, and 3. These sub-interfaces are formed by JR1 domains (JR1-A and JR1-B), JR2 domains (JR2-A and JR2-B), and the TMDs (TMD-A and TMD-B) of each protomer (A & B), respectively (Figs 3D, S5D, and S6D). Because the protomers are related by C2 rotational symmetry, the secondary structure elements in chain A at the dimer interface are a mirror image of the same elements from chain B (Fig 3D). SI-1 and SI-2 contribute to most of the interactions that stabilize the SIDT1 dimer and constitute 65% of the dimer interface. SI-1 is formed by the strands β2, β4, β5, and β7 from JR1-A and JR1-B, which are juxtaposed at the dimer interface and are stabilized by a series of electrostatic and hydrophobic interactions contributed primarily by amino acid stretches N90-S105 and F146-M152. A set of three charged residues, E61, R98, and D148, from both protomers form a network of salt bridges that stabilize SI-1. In addition, SI-1 is stabilized by a conserved hydrogen bonding network contributed by R98, S105, and F146 side chains (Figs 3D, S4C, S5D, and S6D). SI-2 is formed by a β4'–β5' hairpin, which also stabilizes JR-1 and JR-2 interaction (Fig 3A and D). β4'–β5' hairpins from JR2-A and JR2-B are held together by conserved aromatic ring stacking interactions between F233 of JR2-A and H229 of JR2-B. N230 of both protomers also contributes to the SI-2 formation (Figs 3D, S4C, S5D, and S6D). We noticed that only a few amino acids—(F454, Y455, Q461, Y466, and S569) from TM2, TM6, and EL2—within each protomer contribute to form a small hydrophobic SI-3 that is buried within the transmembrane region (Figs 3D, S4D, S5D, and S6D). The TM2s from both protomers are curved inward at the center like the surfaces of a double-concave lens, and the cytosolic halves of TM2s move away from each other and the dimer interface (Figs 2C and 3D). However, the local resolution of SI-3, primarily cytosolic half of the TM2 and the TM6, is poor. As a result, we could not model all the side chains in this region (Figs 1 and S3C and D).

### Comparison of the hSIDT1 structure with its homologs

Although we were preparing and revising this communication, other research groups have either published structures of SIDT1 and its homologs or deposited the unpublished structures in the PDB (33, 34, 35, 36, 37). Here, we compare the following ChUP family structures—apo-hSIDT1 at pH 7.5 reported in this study (8V38), hSIDT1–phosphatidic acid (PA) complex at pH 7.5 (8JUL), lipase-inactive E555Q-hSIDT1 at pH 7.5 (8JUN), hSIDT1–cholesterol (CLR) complex at pH 5.5 (8WOT), hSIDT1–sphingomyelin (SPL)–CLR complex at pH 7.5 (8WOR), apo-hSIDT2 at pH 7.5 (7Y63), apo-hSIDT2 at pH 5.5 (7Y69), hSIDT2 in the presence of RNA at pH 5.5 (7Y68), C. elegans apo-SID (cSID1) at pH 7.5 (8HIP), and cSID1-RNA complex at pH 7.5

**Figure 2. Topology and architecture of hSIDT1.**
**(A)** 2D topology of the ChUP family of proteins. Transmembrane helices (TMs) that form the LBD are highlighted in a gray box, the TMs that form the TMD core in a blue box, and the ECD in a yellow box. The ECD forms a double jelly roll where each jelly roll (JR1 and JR2) is a β-sandwich made of two 4-stranded antiparallel β-sheets. The cytosolic loops (CLs) and the extra-cytosolic loops (ELs) are numbered based on the primary structure. **(B)** TMD dimer is depicted as a cartoon, showing the TM arrangement as seen from extra-cytosolic side and the cytosol. The star indicates the metal ion binding site. **(C)** Two side views of hSIDT1 highlight the dimer interface and ECD-TMD interface. Chain B is displayed as a surface, and chain A is shown as a cartoon with β-strands and TMs colored as they are in the topology diagram. The twofold rotation axis is displayed as a dashed line with an ellipsoid.

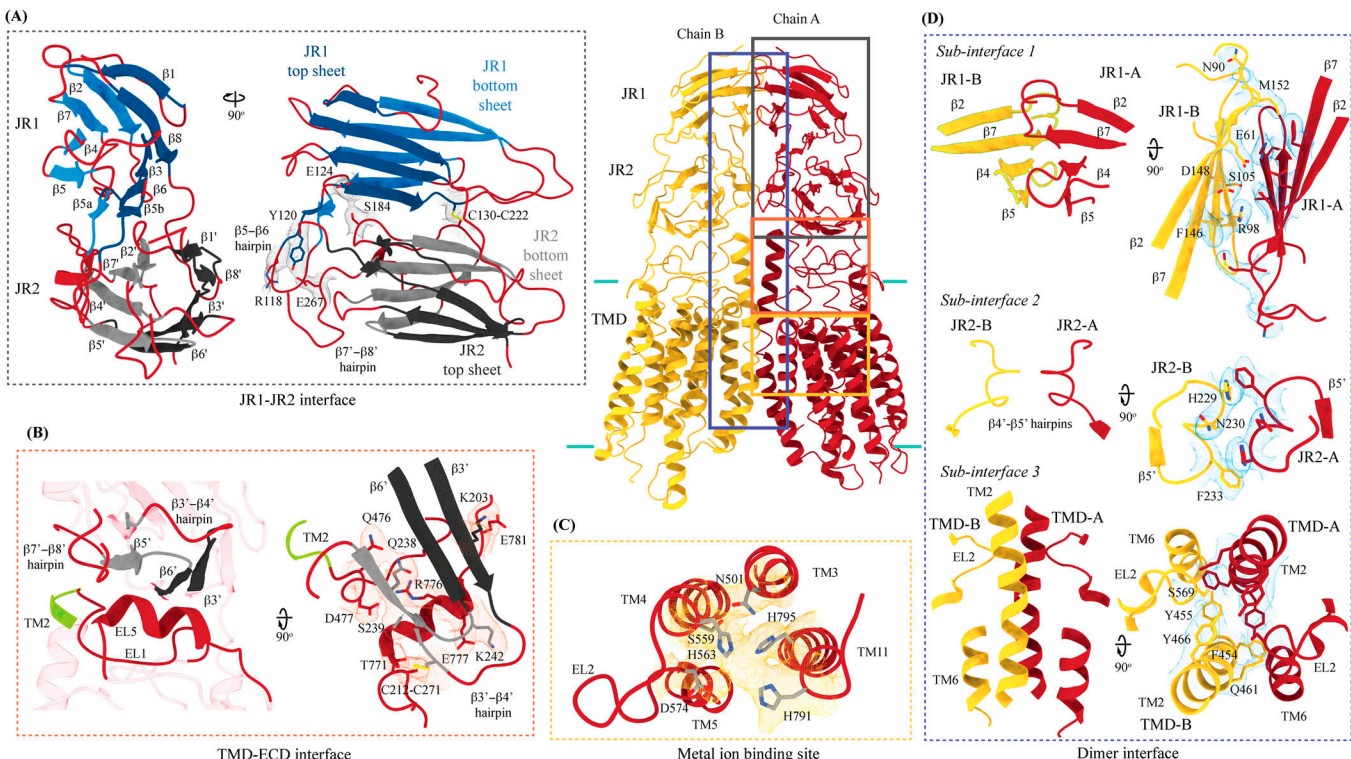

**Figure 3. Inter-chain and intra-chain interactions of hSIDT1.**
Chains A and B are displayed in red and yellow, respectively, at the center. The lipid bilayer is highlighted as cyan bars. **(A, B, C, D)** Regions highlighted in the panels (A, B, C, D) have been indicated on the overall structure with boxes—gray for the JR1-JR2 interface, yellow for the metal ion binding site, orange for the ECD-TMD interface, and blue for the dimer interface. The densities for interactions have been displayed in colors that match their inset highlights (isosurface threshold level of 0.25 for the ECD region and 0.1 for the TMD region in ChimeraX). **(A)** Double jelly roll arrangement of the ECD and the interactions that stabilize JR1 atop JR2. **(B)** Interaction of JR2 with the ECD face of the TMD core allows the ECD to seclude the TMD core from the extra-cytosolic space. **(C)** Ensconced within the TMD core is the evolutionarily conserved metal ion binding site, which is highlighted in yellow. Side chains of amino acids that carve the metal coordination site have been highlighted. **(D)** Three sub-interfaces of the dimer interface, each made by JR1, JR2, and TMD, respectively, have been highlighted in the blue inset.

(8HKE). Our structure comparison analyses delineate inherent flexibility in this family of proteins, especially within the LBD region (Figs 5 and 6 and S7, and Table S2). The TMD core is the most conserved part of the protein and is also stabilized by four tiers of interactions (Figs 3 and 4B and S8C). Thus, we superposed the TMD core of chain A of the dimeric structures of the ChUP family of proteins onto the TMD core of the hSIDT1 structure presented here (RCSB PDB: 8V38) and analyzed the Cα RMSDs of the superposition to study intra-chain flexibility (Figs 5 and S7 and Table S2). Our chain A structure superposition analysis suggests that the LBD is the most dynamic region of the protein, as we see varying degrees of motion in the LBD in all the structures compared with ours. We noticed a significant motion in the LBD of the AlphaFold model of hSIDT1 too when compared to the cryo-EM structure (Figs 5E and S8D). The LBD position in our structure is most similar to the LBD in 8JUN (lipase-inactive mutant) followed by 8WOT (CLR-bound hSIDT1 at pH 5.5), with the exception of TM8 (Fig 5B and D). The LBD motion is relatively similar in 8WOR (SPL-CLR–bound hSIDT1 at pH 7.5) and 8JUL (PA-bound hSIDT1 at pH 7.5) (Fig 5A and C). Incidentally, all hSIDT1 structures included in the comparison, except 8V38 (this study), were obtained in the presence of CHS (Table S1). Perhaps, CHS or cholesterol binding by itself does not alter the LBD dynamics as binding of phospholipids (SPL or PA) at the LBD does. Similarly, the

LBD flexibility appears to be independent of change in pH or RNA binding at the ECD (Figs 5 and S7A–C). Based on the cSID1-RNA complex structure (8HKE), the interface of JR1 and JR2, mainly β5-β6 hairpin and β7′-β8′ hairpin, seems to be the primary site of RNA interaction. We notice movement in this region in our structure, compared with the cSID1-RNA complex structure (Fig S7E and Table S2). JR1 seems to be the second most dynamic region after the LBD across different homologs of the ChUP family, with significant movement in the cSID1 structure (Fig S7). Between cSID1 and hSIDT1 structures, of all the helices, TM2, the primary helix at the dimer interface, is most dynamic (Fig S7D and E).

To study the inter-chain flexibility among the structures, we superposed full chain A or ECD-A or LBD-A of the structures being compared and measured the distances of center of masses (COMs) of these regions in chain B from the superposed COM of these regions in chain A (Fig 6). We noticed that in apo-hSIDT1, the whole chain COMs are ~35 Å from each other, and the ECD COMs are ~28 Å apart, highest among all the hSIDT1 structures (Fig 6A–C). The distance between full chain COMs of all the compared structures is relatively similar, with only up to a ~3 Å difference. However, chain B moves more, perpendicular to the plane of the membrane, in reference to chain A (Fig 6C). For example, this difference is ~9.7 Å between the hSIDT1 structures and the cSID1 structures (Fig 6C).

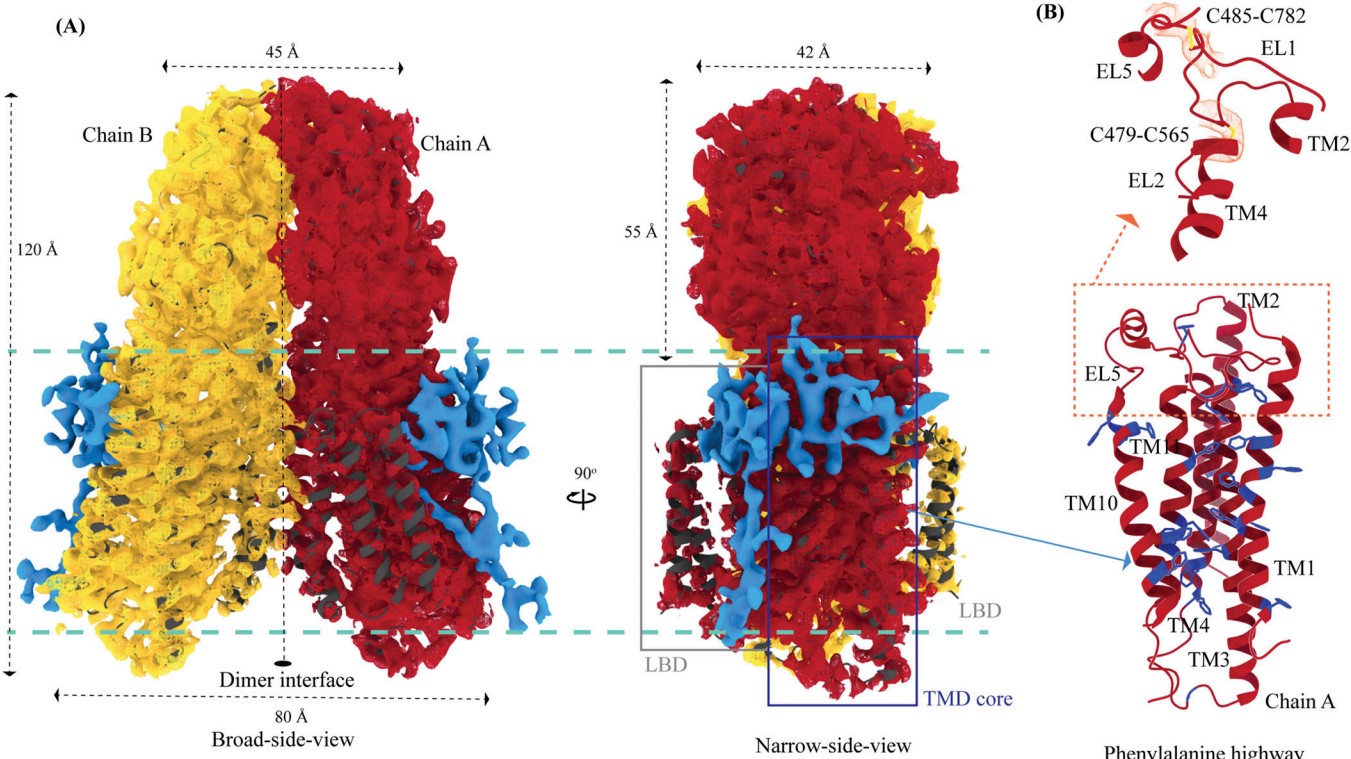

**Figure 4. Potential lipid binding sites.**
**(A)** Ordered lipid or detergent density observed in the final cryo-EM map is displayed in blue, and density for chains A and B is shown in red and yellow, respectively (isosurface threshold level of 0.12 of the C2 refine map as mesh in ChimeraX). We have not modeled either lipid or digitonin in the blue density because we observe relatively poor local resolution for LBD and the LBD-TMD core interface. **(B)** TMD core harbors the phenylalanine highway and metal ion binding site. Highlighted in blue are the phenylalanine residues, primarily from TM11 and TM1, that indicate the presence of phenylalanine highway motif in hSIDT1, as observed in other cholesterol transporters such as ABCG1, NPC1, and PTCH1. The dashed orange inset highlights the disulfides that stabilize EL1, EL2, and EL5 of the TMD core toward the ECD-TMD interface.

We notice a similar vertical movement of ECD-B in reference to ECD-A for the cSID1 structures compared with hSIDT1 structures. In conjunction, we also notice that the difference in the distance between COMs of ECD-B and ECD-A in the cSID1 structures compared with hSIDT1 structures is almost twice (~7 Å) the difference in distance between full-length chain A and B COMs (~3 Å). These observations suggest a difference in the directionality of motion between the TMDs and ECDs in one protomer in reference to the other protomer in a dimer.

A noteworthy observation is the position of COMs of the ECD regions and full chains of all the hSIDT1 structures that are compared (Fig 6B and C). Although the ECD positions remain relatively unchanged, chain B moves down more with respect to chain A in E555Q mutant and cholesterol-bound structures than other hSIDT1 structures, suggesting a motion in the TMD-B region (Figs 5 and 6C and S5D). To probe this further, we compared intra-chain and inter-chain LBD dynamics between the available hSIDT1 structures (Fig 6D and E). The LBD-A of apo-hSIDT1 superposed well with the LBD-A of the E555Q-hSIDT1 and hSIDT1-CLR complex at pH 5.5, but not with the LBD-A of the hSIDT1-PA complex at pH 7.5 or the hSIDT1-SPL-CLR complex at pH 7.5 (Fig 6D). However, the LBD-As of the hSIDT1-PA complex at pH 7.5 and the hSIDT1-SPL-CLR complex at pH 7.5 superposed partially on each other, with more movement in TMs5–7.

Within the LBD-As of various hSIDT1 structures, most dynamic regions appeared to be TMs5–7 (Figs 5 and 6D and S5D). Next, we compared LBD-B motion in reference to superposed LBD-As and noticed that the LBD-B moves significantly along the plane of the membrane in reference to the LBD-A. For example, the LBD-B of the hSIDT1-SPL-CLR complex at pH 7.5 is situated ~22.7 Å away along the plane of the membrane from the LBD-B of apo-hSIDT1 at pH 7.5. A similar LBD-B position is also seen in the hSIDT1-PA complex at pH 7.5. However, the LBD-Bs of the E555Q-hSIDT1 and the hSIDT1-CLR complex all at pH 5.5 do not move with respect to the LBD-B of apo-hSIDT1 at pH 7.5, suggesting that neither cholesterol binding nor change in pH causes LBD motion. Perhaps binding of phospholipids such as PA or SPL at the LBD alters the LBD-TMD core interface, allowing for independent TMD core and LBD movements enabling dsRNA transport.

## Discussion

siRNA-mediated RNAi begins with the poorly understood mechanism of transport of small non-coding dsRNA into a cell. Once inside the cell, the dsRNA is processed in a manner that is conserved across different kingdoms of life. Briefly, the incoming

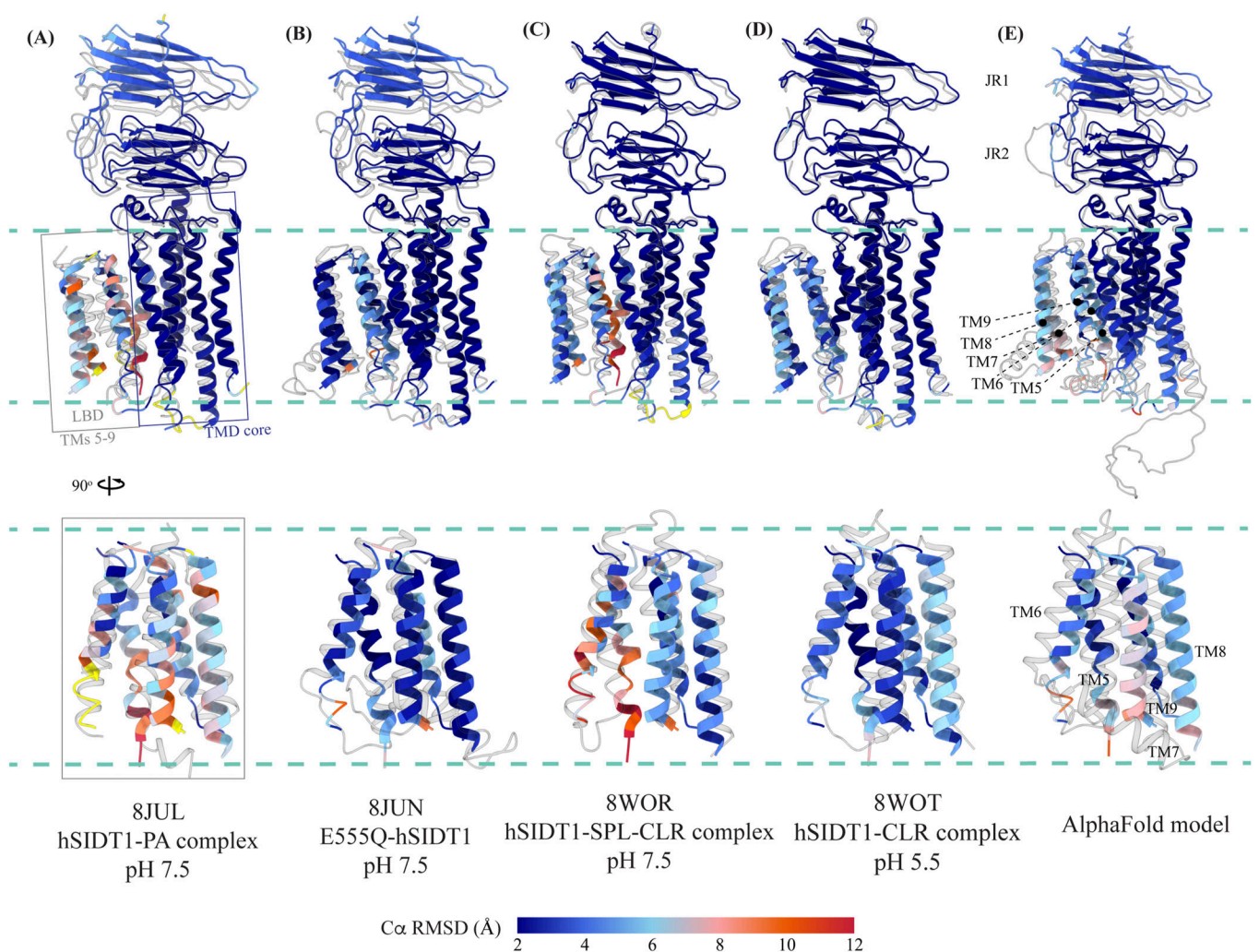

**Figure 5.  Comparison of intra-chain dynamics in various hSIDT1 structures.**
We used sequence-based pairwise alignment in ChimeraX to color hSIDT1 structure in this study using the Cα RMSD obtained upon aligning it with various structures (Table S2). The TMD core of chain A from all the structures was used for alignments. The color key denotes the range of Cα RMSD of the pairwise alignments, with blue for the lowest RMSD, red for the highest, and yellow for regions missing from the alignment. **(A, B, C, D)** Each of the panels shows chain A of apo-hSIDT1 (8V38) colored based on the RMSDs of alignments with the (A) hSIDT1-PA complex at pH 7.5 (8JUL), (B) E555Q-hSIDT1 at pH 7.5 (8JUN), (C) hSIDT1-SPL-CLR complex at pH 7.5 (8WOR), and (D) hSIDT1-CLR complex at pH 5.5 (8WOT). **(E)** shows superposition with the AlphaFold model of hSIDT1 (33, 34). All the structures used for superposition with apo-hSIDT1 (8V38) have been displayed in gray with their respective PDB IDs indicated below each panel. Based on the alignments, the LBD dynamics do not appear to be determined by CLR binding or change in pH.

dsRNA is trimmed to 21- to 23-bp short interfering RNA by the RNase III family of enzymes. The processed dsRNA is unwound by an RNA-induced silencing complex that mediates the hybridization of the single-stranded antisense RNA with target mRNA and eventual degradation of the homologous mRNA (52, 53). A large portion of the existing knowledge on the complicated specific cellular transport of small non-coding RNAs has been obtained by studying RNAi in *C. elegans* (2, 11, 12). Human homologs of the ChUP family of proteins—hSIDT1 and hSIDT2—are believed to be lipophilic dsRNA transporters (17, 18, 29). However, the exact physiological role of SIDTs has been a longstanding debate, with various activities such as the transport of cations, cholesterol, DNA, ATP-dependent protein and peptides, and, most recently, lipid hydrolysis being attributed to them in the literature (18, 24, 25, 29, 54, 55, 56). The

emerging consensus on the function is that SIDTs are cholesterol-dependent dsRNA transporters with a divalent metal ion–dependent lipase activity within the transmembrane region (29, 33, 34, 35, 36, 37, 57).

While preparing this article, other research groups reported the structural and functional characterization of recombinant human SIDTs (33, 34, 35, 36, 37). All these reports have differences in the sample preparation protocol that we have compiled in Table S1. At the time of this comparison, the hSIDT1 PDBs from the Zheng et al. and Hirano et al. studies were unreleased. Hence, we could not discuss the effect of differences in sample preparation on the hSIDT1 structures from these studies. The Qian et al, Liu et al, and Sun et al studies used HEK293F cells to produce full-length hSIDT2 and hSIDT1 (33, 34, 37). All three of these studies also use cholesteryl

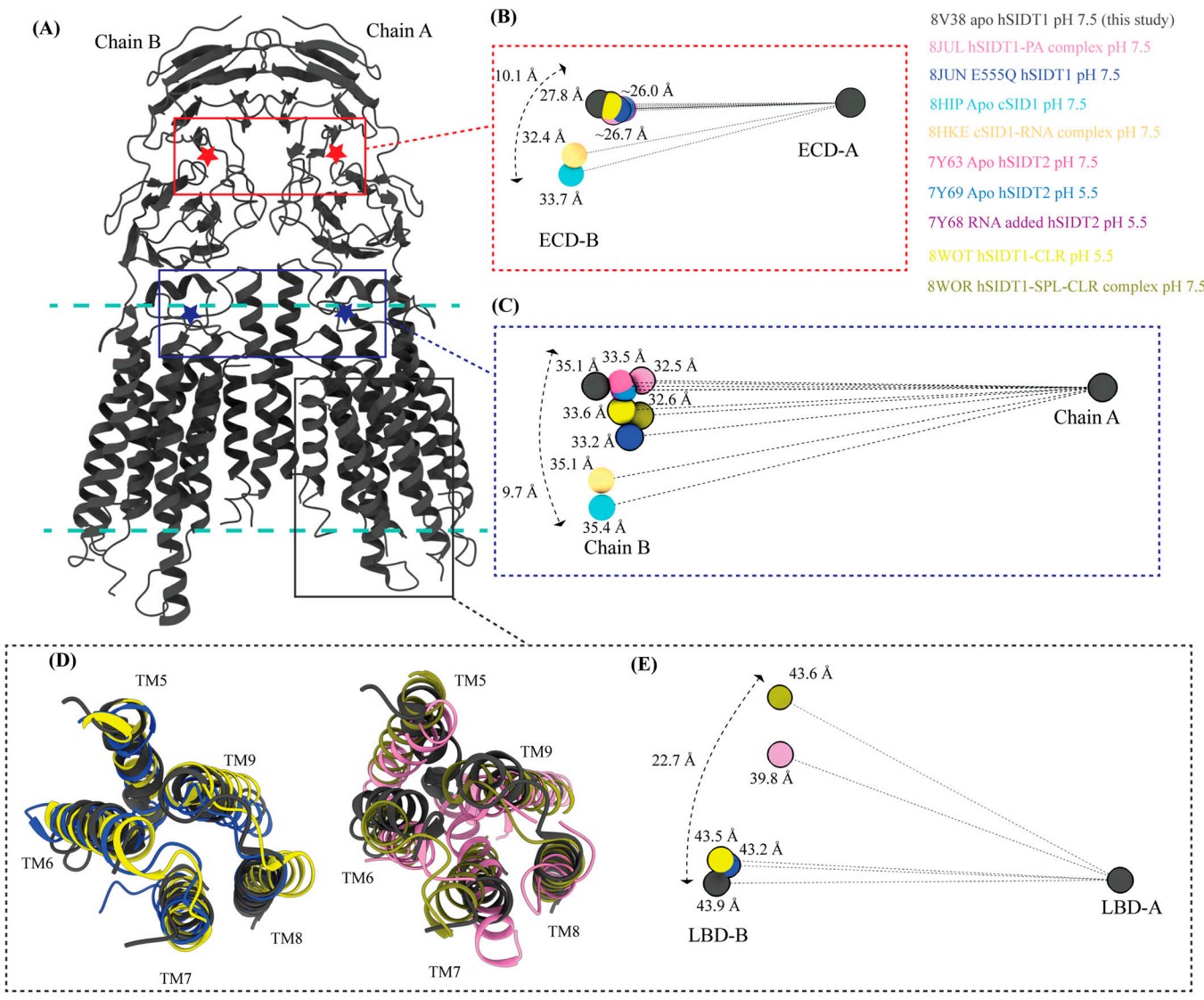

**Figure 6. Inter-chain dynamics in hSIDT1.**
**(A)** hSIDT1 dimer (gray) with ECD (red star) center of masses (COMs) and full chain COMs (blue star) highlighted by red and blue boxes. **(B, C, E)** Distances on the spheres in panels (B, C, E) denote the distances of COMs of the regions being compared in chain B (ECD-B and LBD-B) from COMs of that region in superposed chain A (ECD-A and LBD-A). The spheres representing all SIDT1 structures have been highlighted with a black border. The double-ended arrow denotes the distance chain B COM of that region has moved relative to 8V38 (gray sphere) chain B COM, when chain As of the structures were superposed. **(B)** Relative position of the COMs of chain B ECDs of the homologs compared with hSIDT1 from this study (8V38; gray sphere). The relative positions of these COMs were calculated by superposing the chain A ECDs of all the homologs. **(C)** Relative position of the COMs of the entire chain B of the homologs compared with COM of chain B of 8V38 was calculated by superposing the chain A of all the homologs. In panels (B, C, E), only 8V38 chain A COM is displayed for clarity. **(D, E)** are ECD views of LBDs of various hSIDT1 structures. In panel (D), we see superpositions of the LBDs of chain A of various hSIDT1 structures. In panel (E), the center of masses of LBD regions of chain B (LBD-B) of hSIDT1 structures is compared upon superposition of the LBD of chain A (LBD-A). Relative to the LBD-A, the phospholipid-bound hSIDT1 structures, 8JUL and 8WOR, show the highest outward motion from the plane of the dimer.

hemisuccinate (CHS) during solubilization and purify the protein in glyco-diosgenin (GDN) for final sample vitrification. All three groups maintained a Vitrobot chamber at a temperature of 4–8°C for sample blotting. We used HEK293S GnTI⁻ cells for protein expression, excluded CHS during solubilization, and purified the protein in digitonin for final sample vitrification at a sample blotting temperature of 18°C. As a result, our sample is apo-hSIDT1 that lacks complex *N*-glycosylation and is vitrified at higher temperature and pH. Lack of hSIDT1 ligands, such as RNA or cholesterol or

phospholipids or sphingolipids in the sample preparation, combined with the differences in the grid vitrification conditions might have introduced greater heterogeneity in our cryo-EM sample leading to the differences that we see in our structure compared with the ChUP family member structures reported by other groups (Table S1).

The ECD and the disordered cytosolic domain (CD) connecting TM1 and TM2 have been shown to bind to RNA in a pH-dependent manner, with more efficient binding at low pH (31, 32, 33, 34, 35, 36,

37, 38, 58). In hSIDT2, even though the cryo-EM sample at low pH contained RNA, it could not be observed in the cryo-EM map. However, in the presence of RNA, an unexplained density appeared toward the cytosolic side—perhaps a better resolved CD (33). In hSIDT1, deletion of CD resulted in an expression construct with diminished RNA binding ability (35). Mendez-Acevedo et al. demonstrated that SIDTs primarily transport cholesterol, and they transport RNA only upon being conjugated to cholesterol (29). SIDTs are believed to have three CRAC motifs (L/V-X$_{[1-5]}$-Y/F/W-X$_{[1-5]}$-R/K). In hSIDT1, they are located between L155-K165 ($\beta$8 of JR1), L584-R593 (TM5), and L643-R652 (TM7). In the unpublished RNA-bound cSID1 structure, RNA has been modeled at the JR1-JR2 interface, closer to the first CRAC motif in JR1 (8HKE). However, Liu et al. model two cholesterol sites between TM8 and TM10 and between TM2 and JR2, neither of which are canonical CRAC motifs. In all the structures of the ChUP family, phospholipid interactions have been observed at the LBD or LBD-TMD core interface, and the LBD is believed to possess lipase activity (33, 34, 35, 36, 37). Hirano et al. suggest that hSIDT1 lipase activity has been observed in the presence of methyl-$\beta$-cyclodextrin, emphasizing that removing cholesterol enhances the lipid hydrolysis (35). Perhaps cholesterol acts as an allosteric regulator of lipase activity. All the previously reported hSIDT1 structures have been obtained in the presence of cholesterol or phospholipid or lipase-inactive E555Q mutation or low pH (33, 34, 35, 36, 37). In our experimentally determined hSIDT1 structure, we did not use cholesterol or its analogs in our sample preparation and conducted our study at high pH. Although we see density for ordered lipids/detergents in our structure, they are all predominantly located at the extra-cytosolic side of the phenylalanine highway (Fig 4). Hence, the LBD in our structure is poorly resolved than the other hSIDT1 structures. Moreover, our structure comparison analyses suggest that the most flexible across all hSIDT1 structures is the LBD, especially TMs5–7 (Fig S7F).

Grishin and colleagues, using sensitive sequence similarity searches, classified a group of putative transmembrane receptors and transporters into a superfamily called CREST (alkaline ceramidase, PAQ receptor, Per1, SID-1, and TMEM8) (59). The CREST superfamily of membrane proteins comprises metal-dependent lipid-modifying hydrolases that possess a conserved S-H-D-H-H metal ion coordination site formed by the juxtaposition of three TMs. These residues are conserved in the TMD core of the ChUP family of proteins (Figs S6C and S8C). All the available ChUP family structure reports model a zinc ion at this coordination site, and the lipase activity of SIDT-like proteins is zinc-dependent (33, 34, 35, 36, 37). In our hSIDT1 structure, we find unexplained density in this site carved by S559, H563, D574, H791, and H795 (Figs 3C and S5C). Because of a nominal resolution of ~3.4 Å, and because of a lack of biochemical evidence for zinc in our samples, we left this site unmodeled. H795F mutation, like E555Q, in hSIDT1, reduces the lipid hydrolysis activity (34). Also, the analogous mutations S559I and H791Y in *C. elegans* attenuate systemic RNAi (58). These observations show that a metal ion is not only essential to maintain the TMD core integrity but also crucial for lipase hydrolysis and RNA transport. We also notice that the amino acids that upon being mutated in *C. elegans* abrogate the systemic RNAi, and are conserved in the ChUP family of proteins, all predominantly lie in the vicinity of the path traced by the cavity in our MOLE analysis (Fig S8B

and C) (35, 58, 60). P109S mutation in the $\beta$5-$\beta$6 hairpin, at the JR1-JR2 interface, lies near the RNA binding site as per the RNA-bound cSID1 structure (8HKE). C479Y at the TMD-ECD interface is a part of the C479–C565 disulfide essential for stabilizing the TMD core toward the extra-cytosolic side. C479Y mutation breaks this disulfide and, as a result, could destabilize the TMD-ECD interface (Figs 3B and 4B). R593C mutation could destabilize the hinge region at the LBD-TMD core interface (Fig S8C). Four other conserved residues that abrogate systemic RNAi are G503, G508, S559, and H791. All these residues are situated in the TMD core with S559 and H791 being a part of metal ion coordination, and drastic mutation here potentially affects the stability of the TMD core, lipase activity, and subsequently protein function (Figs 3C and S8C).

The exposed face of the ECD is covered with basic amino acids conducive to nucleic acid binding, especially at the interface of JR1 and JR2 (Fig S8A). In fact, in the RNA-bound cSID1 structure (8HKE) the RNA has been modeled at the JR1-JR2 interface. The MOLE cavity analysis of the hSIDT1⁻ structure revealed the presence of two cavities in tandem, each of about 110–120 Å in length, which extend from the JR1-JR2 interface to the cytosolic opening of the protomer passing through the LBD-TMD core interface (Fig S8B). In our structure comparison analysis, we observe that LBD is the most dynamic part of the ChUP family of proteins followed by JR1 and that the motion in the ECD is prominent in the structures where RNA is bound at the JR1-JR2 interface. We also notice that phospholipid binding at the lipase site causes the LBDs to move away from the TMD core (Figs 5 and S7). We presume this LBD motion induced by binding of lipids, such as PA or SPL, is essential for the lipid hydrolysis activity. Although it has been suggested that RNA transport is dependent on zinc-dependent lipase activity, which in turn is induced by removal of cholesterol, the mechanism by which lipid hydrolysis aids the RNA that is bound at the ECD traverse TMD is unclear. A combination of single-molecule FRET, time-resolved cryo-EM, and molecular dynamics studies of apo-hSIDT1 in the presence of RNA and lipids could provide more insights into the role of lipid-induced LBD motion in transport. Previous reports involving studies of ECDs suggested that ChUP family proteins form tetramers (31, 32). However, all the available structures of ChUP family members are dimers. We believe that further structural characterization of SIDT1, with a specific focus on connecting the roles of RNA binding at the ECD, metal-dependent lipase activity within the TMD, and subsequent LBD dynamics in a tetramer, will provide us with molecular snapshots of dsRNA transport.

# Materials and Methods

### Cell culturing and protein expression

The codon-optimized gene encoding full-length human systemic RNAi defective transmembrane family member 1 (hSIDT1) was synthesized by GenScript and was then cloned into the pEG BacMam expression vector to be expressed via baculoviral transduction in HEK293S GnTI⁻ cells as a fusion protein containing a C-terminal GFP-Strep-tag-II for large-scale protein expression (61). For small-scale detergent screening, adherent HEK293S GnTI⁻ cells

were cultured in DMEM with 10% FBS at 37°C. Before transfection, the cells were washed with 1X PBS and were supplied with fresh prewarmed DMEM containing 10% FBS. Cells were transfected at 80% confluency. ~1 × 10⁶ cells were transfected with 1 $\mu$g DNA using TurboFect (Thermo Fisher Scientific), in serum-free DMEM, as per the manufacturer's protocol. The cells expressing hSIDT1 were grown at 37°C and 5% $CO_2$ for 8–10 h, after which the culture media were replaced with fresh prewarmed DMEM containing 10% FBS and 10 mM of sodium butyrate. Subsequently, the cells were grown at 32°C and 5% $CO_2$ for an additional 24–36 h, before being harvested.

Baculovirus preparation was done in Sf9 cells grown in Sf-900 III serum-free medium. Briefly, DH10Bac cells (Thermo Fisher Scientific) were transformed with a hSIDT1 expression vector, and lacZ⁻ colonies were selected for bacmid DNA isolation. 1 × 10⁶ Sf9 cells were transfected with 1 $\mu$g of DNA using Cellfectin II reagent as per the manufacturer's protocol. Transfected cells were grown at 27°C for 96 h. The supernatant from the cell culture was harvested, filtered through a 0.2-$\mu$m filter, and used as P1 virus. 100 $\mu$l of P1 virus was added to 1 liter of Sf9 cells at a cell density of 1 × 10⁶/ml in serum-free Sf-900 III media. The transduced cells were grown for 96 h at 27°C while shaking at 120 rpm, and spun down at 4,000$g$ for 20 min, and the supernatant media were filtered through a 0.2-$\mu$m filter and used as P2 baculovirus. For the large-scale expression of hSIDT1, baculoviral transduction of HEK293S GnTI⁻ cells was performed in FreeStyle 293 expression media containing 2% FBS. Mammalian cells at a density of 2.5 × 10⁶ cells/ml were transduced using P2 virus at a multiplicity of infection of 1.5–2 and were incubated on an orbital shaker at 37°C and 5% $CO_2$ for 8–10 h. After this, the cells were supplemented with 10 mM sodium butyrate and were incubated in a shaker at 32°C and 5% $CO_2$ for an additional 38–40 h. The harvested cell pellet was stored at –80°C until further use.

### Purification of hSIDT1

The sonicated lysate of HEK293S GnTI⁻ cells expressing *C*-terminal GFP fusion of hSIDT1-FL was subjected to ultra-centrifugation at 185,000$g$ for 1 h at 4°C to harvest membranes. Membranes were resuspended using a Dounce homogenizer in 25 mM Tris–HCl, pH 7.5, 200 mM NaCl, 1 mM PMSF, 0.8 $\mu$M aprotinin, 2 $\mu$g/ml leupeptin, and 2 $\mu$M pepstatin A. To this membrane suspension, an equal volume of 2% digitonin solution prepared in membrane suspension buffer was added. The membrane suspension was solubilized at 4°C for 90 min. The solubilized membrane suspension was centrifuged at 185,000$g$ for 1 h at 4°C. 2 ml of Strep-Tactin affinity resin was packed into a column, per every liter of cell culture, and the solubilized supernatant was passed through it at a flow rate of ~0.3–0.5 ml/min. Affinity resin saturated with hSIDT1-FL was washed with 5-6 column volumes of 0.5% digitonin, 25 mM Tris–HCl, pH 7.5, and 200 mM NaCl. The bound protein was eluted using 5 mM D-desthiobiotin prepared in the wash buffer. The homogeneity of purified hSIDT1 was confirmed by performing FSEC (62). We confirmed the purity of hSIDT1 by SDS–PAGE and peptide mass fingerprinting. The elution fractions containing homogeneous and pure protein were pooled and concentrated to ~2 mg/ml. hSIDT1 was further purified by size-exclusion chromatography using Superdex S200 pre-equilibrated with 0.5% digitonin, 25 mM

Tris–HCl, pH 7.5, and 200 mM NaCl. Fractions corresponding to the dimer hSIDT1 peak were analyzed for their purity and homogeneity on SDS–PAGE and FSEC and were concentrated to 1 mg/ml (Fig S1F and G). The concentrated protein was incubated with 100 $\mu$M fluorinated octyl-maltoside (FOM) for 20-30 min on ice, and centrifuged at 18,000$g$ for 15 min, and the supernatant was used for preparing cryo-EM grids (Fig S1H).

### Detergent screening by fluorescence-detection size-exclusion chromatography

Briefly, 100,000 transiently transfected cells expressing hSIDT1-GFP were solubilized in 100 $\mu$l of 1% of detergent prepared in 25 mM Tris–HCl, pH 7.5, 200 mM NaCl, 1 mM PMSF, 0.8 $\mu$M aprotinin, 2 $\mu$g/ml leupeptin, and 2 $\mu$M pepstatin A. The solubilized lysate was centrifuged at 185,000$g$ for 1 h at 4 °C, and the supernatant was analyzed by FSEC on a Superose 6 Increase 10/300 GL column pre-equilibrated with 0.15 mM LMNG, 25 mM Tris–HCl, pH 7.5, and 200 mM NaCl (Fig S1A–E). To evaluate the quality of the purified protein during the protein purification process, and to estimate the molecular weight of the dimer, ~10 $\mu$l of elution fractions was analyzed in a similar fashion (Fig S1).

### Cryo-EM sample preparation and data collection

UltrAuFoil holey-gold 300-mesh 1.2/1.3-$\mu$m size/hole space grids (UltrAuFoil, Quantifoil) were glow-discharged, using a PELCO glow discharger, for 1 min at 15 mA current and 0.26 mBar air pressure before sample vitrification. 2.5 $\mu$l of hSIDT1 at 1 mg/ml was applied to the glow-discharged grids, and subsequently blotted for 2.5 s at 18°C and 100% humidity using a Vitrobot (mark IV; Thermo Fisher Scientific). The grids were plunge-frozen in liquid ethane, without any wait time. Movies were recorded on a Gatan K3 direct electron detector in super-resolution counting mode with a binned pixel size of 0.85 Å per pixel using SerialEM on a FEI Titan Krios G4i transmission electron microscope (Thermo Fisher Scientific) operating at an electron beam energy of 300 KeV with a Gatan Image Filter slit width set to 20 eV. Exposures of ~2 s were dose-fractionated into 50 frames, attaining a total dose of ~50 $e^-$ Å$^{-2}$, and the defocus values were varied from –1.2 to –2.5 $\mu$m.

### Cryo-EM data processing

Image processing was done entirely using CryoSPARC (v 4.2.1) (41). Briefly, motion correction was carried out on the raw movies, and subsequently, the contrast transfer function (CTF) was estimated using patch-CTF correction on dose-weighted motion-corrected averages within CryoSPARC (Fig S2). A total of 20,764 micrographs were collected, of which 15,391 were selected for further processing. The elliptical Gaussian blob picker was used to pick 3,204,173 particles. Particles were extracted initially using a box size of 336 pixels. A subset of the particles picked by the blob picker was used to generate a low-resolution ab initio reconstruction, which was later used as a reference for heterogeneous refinement. The particles were subjected to multiple rounds of reference-free 2D classification and heterogeneous refinement to

get a cleaned subset of particles, which resulted in classes with recognizable features. The cleaned particles were re-extracted with a box size of 360 pixels, resulting in the particle stack of 302,224 particles that were further processed stringently by heterogeneous refinement and 2D classification. Finally, only the particles belonging to micrographs with a CTF fit <4 Å were selected for subsequent refinements. A final subset of 122,683 particles was used for NU refinement, yielding a 3.5 Å map (Fourier shell coefficient = 0.143 criterion) (42). Local refinement with a focused mask on the entire protein was performed to improve local density, resulting in a final map of 3.4 Å. Both sharpened and unsharpened maps were used to finalize the fit of the side chains during model building and refinement (Fig S3D). The final 3D model was validated using the sharpened map for deposition, and local resolutions were estimated on the sharpened map using RESMAP software (63).

### Model building, refinement, and structural analysis

ModelAngelo was used to build the hSIDT1 ECD into the final map (45). The AlphaFold hSIDT1 model (UniProt ID: Q9NXL6) was used to build the TMD core (46, 64). The LBD was built manually, and the entire structure was analyzed for the Ramachandran and geometry outliers in Coot (47). The final model refinement was performed in Phenix (43). We used sequence-based pairwise alignment in ChimeraX to perform the superpositions of the TMD core of chain A of hSIDT1 and its homologs (44). We interpreted the results of alignments by monitoring either the resultant C$\alpha$ RMSD or the change in the center of mass of a particular domain upon alignment. ChimeraX was also used for making the figures. RNA transport channel prediction on hSIDT1 was performed using the MOLEonline server (50).

## Data Availability

The correspondence for material and data presented in this article should be addressed to navratna@umich.edu or mosalaga@umich.edu. The cryo-EM maps and 3D coordinates of human SIDT1, along with the half-maps and the masks used for refinement, have been deposited under the accession codes EMD-42943 and PDB-8V38 in the Electron Microscopy Data Bank (EMDB) and Protein Data Bank (PDB).

## Supplementary Information

## Acknowledgements

We thank the University of Michigan (UM) cryo-EM facility staff for assistance with cryo-EM data collection, and UM BSI and UM LSI for generously supporting the UM cryo-EM facility. All members of the Mosalaganti laboratory and the Baldridge laboratory, and Dr. Jonathan Coleman, Dr. Farzad Jalali-Yazdi, and Dr. Prashant Rao are acknowledged for their helpful comments on our work. The research presented here was funded by the Klatskin-Sutker Discovery Fund award to V Navratna and the start-up funding provided to S Mosalaganti by the University of Michigan. S Mosalaganti is also a recipient of the NIH Director's New Innovator Award (DP2) (1DP2GM150019-01).

## Author Contributions

V Navratna: conceptualization, data curation, formal analysis, supervision, funding acquisition, validation, investigation, visualization, methodology, project administration, and writing—original draft, review, and editing.
A Kumar: formal analysis, validation, investigation, visualization, and writing—review and editing.
JK Rana: data curation, validation, and writing—review and editing.
S Mosalaganti: conceptualization, supervision, funding acquisition, visualization, project administration, and writing—review and editing.

## Conflict of Interest Statement

The authors declare that they have no conflict of interest.

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
