## [Reviewer comments · Life Science Alliance]

Life Science Alliance

Structure of the human SIDT1 reveals the conformational flexibility of its lipid binding domain

Vikas Navratna, Arvind Kumar, Jaimin Rana, and Shyamal Mosalaganti

DOI: <https://doi.org/10.26508/lsa.202402624>

Corresponding author(s): *Shyamal Mosalaganti, University of Michigan-Ann Arbor and Vikas Navratna, University of Michigan*

Review Timeline:

Submission Date:	2024-01-26
Editorial Decision:	2024-03-18
Revision Received:	2024-04-30
Editorial Decision:	2024-05-31
Revision Received:	2024-06-10
Accepted:	2024-06-12

Transaction Report:

March 18, 2024

Re: Life Science Alliance manuscript #LSA-2024-02624-T

Dr. Shyamal Mosalaganti
University of Michigan
Life Sciences Institute
Mary Sue Coleman Hall
210 Washtenaw Avenue
Ann Arbor, MI 48109

Dear Dr. Mosalaganti,

Thank you for submitting your manuscript entitled "Structure of the human SIDT1 reveals the conformational flexibility of its lipid binding domain". The manuscript has been evaluated by expert reviewers, whose reports are appended below. Unfortunately, after an assessment of the reviewer feedback, our editorial decision is against publication in Life Science Alliance.

Although your manuscript is intriguing, I feel that the points raised by the reviewers are more substantial than can be addressed in a typical revision period. If you wish to expedite publication of the current data, it may be best to pursue publication at another journal.

Given the interest in the topic, I would be open to re-submission to Life Science Alliance of a significantly revised and extended manuscript that fully addresses the reviewers' concerns and is subject to further peer review. If you would like to resubmit this work to Life Science Alliance, you may submit an appeal directly through our manuscript submission system. Please note that priority and novelty would be reassessed at re-submission.

Regardless of how you choose to proceed, we hope that the comments below will prove constructive as your work progresses.

Thank you for thinking of Life Science Alliance as an appropriate place to publish your work.

Sincerely,

Reviewer #1 (Comments to the Authors (Required)):

Summary

Navratna, Kumar et al. report a cryo-EM structure of the detergent-solubilized human SIDT1 in a cholesterol-free environment. In *C. elegans*, SID1 is crucial for the uptake of short non-coding RNA from the environment, leading to systemic RNA interference. These proteins are conserved in humans, despite the process of systemic RNAi not being conserved. The authors meticulously describe the general architecture of hSIDT1 and the interfaces between the two dimers and the subdomains of hSIDT1. Considering the solely structural nature of this paper and lack of any functional characterization, the submitted manuscript has limited novelty. Multiple structures of hSIDT1 and hSIDT2 were published within the last 9 months by other groups. The authors take references to these published structures by comparing their apo-structure with the previously deposited structures of hSIDT1, hSIDT2 and *C. elegans* SID1. They identify the lipid-binding domain of hSIDT1 to be flexible in the absence of cholesterol and hypothesise a potential role in lipase activity. Overall, the manuscript is well-written and the figures are accurate, but sometimes too dense with labels.

Major concerns

- The lipid binding domain of cholesterol-free hSIDT1 was described as highly flexible. The authors did not attempt to resolve the flexibility of the lipid-binding domain (LBD) of the transmembrane part of the protein. A global mask was applied during refinement, but did the authors try to refine the structure using a local mask on the LBD? Furthermore, the number of particles should allow for further classification of the particles, using either 3D classification, 3D variability analysis or 3D flex analysis in cryoSPARC or equivalent methods in other processing softwares, in order to better resolve the flexibility of the LBD.
- The authors state that the observed flexible LBD in the cholesterol-free structure facilitates the lipase activity and base this on

the structure comparison with other hSIRT1/hSIRT2 structures. However, the comparison of the structures leads to the conclusion that the LBD of apo hSIRT1 is nearly identical to the LBD of the mutant E555Q, a mutant with abrogated lipase activity. This conclusion seems contradictory, as the facilitation of lipase activity and the abrogated lipase activity are contrary.

- The structure comparison is incomplete. There are two more hSIRT1 structures available on PDB (8KCW and 8KCX), which were not included in the comparison. Could the authors include these two comparisons into Figure 3 and the main text?

Minor concerns

- Title for Figure 2: "Critical inter-chain and intra-chain interactions of hSIRT1" - There is no biochemical evidence for the interactions to be critical. Perhaps a more modest title would be a better fit.
- The panels in Figure 2A and 2D are hard to read due to the many labels.
- Figure S1 - panel F - Could the authors comment on the second band that is visible on the gel?
- Chapter 2.3. Comparison of hSIRT1 structure with its homologs - the authors write "we used TMD core of these proteins to superpose chain A of each structure onto our hSIRT1 map (RCSB PDB: 8V38)" but they are not referring to a map rather to the corresponding model.
- "A noteworthy observation is the position of COMs of the ECD and full chains of the E555Q hSIRT1 mutant (8JUN) in comparison with wild-type hSIRT1 (Fig. 5B and 6C)." - There is no Figure 6C.
- Discussion - the authors state the differences in the purification and vitrification conditions between their work and the other published work, however they do not explain satisfyingly what may cause the differences.
- Table 1 - Validation statistics - the model contains 0.6% Ramachandran Outliers. This needs to be improved to reduce the statistics to 0% Ramachandran outliers, unless the outliers are supported by high resolution density.

Referee Cross-Comments

I agree with the other referees comments.

Reviewer #2 (Comments to the Authors (Required)):

Comments to the Authors:

While I am sure it is disappointing to have other publications emerge as you are working on analysis and publication of your hSIRT1 data, in its current state, this manuscript does not add significant findings to the field. I have made this decision on the grounds that the paper contains only structural data, and that data is not of a high enough quality to support the conclusions made. While I appreciate that the authors have attempted to add novelty by comparing their structure with other recently available structures and performing predictive mutagenesis on the SIRT1 protein, the cryo-EM data produced by the authors was not of high enough quality for these comparisons to be reliable or meaningful.

A short summary of the paper and main points, including description of the advance offered to the field.

This manuscript provides a Cryo-EM map of hSIRT1 protein at 3.4 Å (3.98 Å unmasked), due to the resolution of the map the authors fit a model and determine a structure of the protein hSIRT1. They also compare their SIRT1 structure to others in the field and model some mutations in the protein.

The main novel findings the authors are claiming is:

1) That the lipid binding domain was unable to be resolved because they isolated the protein without cholesterol, and this made the lipid binding domain too flexible to resolve.

It is stated that "TMs 5-9 form a conformationally flexible domain which is poorly resolved in our cryo-EM map" in your LBD results section. This is a vast understatement. These regions aren't poorly resolved, they are actually absent for the most part. It is not possible to model this region of the protein with the quality of the map, and given the major conclusions of the paper surround this region, these conclusions cannot be substantiated from these data.

While the hypothesis the authors make regarding the flexibility of the LBD is reasonable to make, for this claim to be substantiated, the following (significant) steps would need to be addressed:

- 1- Re-analyze the cryo-EM data/ collect additional data and perform further analysis e.g. 3DVA/3D flex.
- 2- Collect data with cholesterol bound hSIRT1 to demonstrate that the lack of cholesterol here is specifically the cause of the low resolution (rather than the quality of the cryo-EM data or stability of the membrane protein in the detergent membrane mimetic of choice).
- 3- If you want to test how this domain moves, you need a better structure and to run molecular dynamics simulations.
- 4- Perform additional experiments to analyze movement of this domain in the presence or absence of cholesterol (e.g. smFRET/molecular dynamics).

2) That some modelled mutations affect the structure of the protein. These mutations should be made, and functionally assessed.

The final statement in the discussion is a major issue:

"we hypothesize that the RNA binding at the JR1-JR2 interface induces a rearrangement of JR1 and JR2 which allows for the formation of an opening that can feed RNA to the TMD. Then, the metal ion dependent lipase activity in the TMD region disturbs the lipid bilayer around the protein and allows for the LBD and TM2 to rearrange, enabling the formation of a cavity large enough to allow the passage of RNA to cytosol."

Again, like the issues of the results, the authors cannot make this prediction with the cryo-EM data at the level of quality presented in this paper or without significant additional experiments (as outlined above) to warrant proposing this hypothesis.

Additional issues to be addressed

1- The second paragraph of the introduction, 3rd sentence is missing an 'a'

1. should be "that is a popular cancer therapeutic"

2- Watch the consistency of you naming of HEK293S GnTi-, pick a name and stick with it

3- The specifications of the machine or server the data was processed on. While unconventional, open sharing of this information should become the gold standard as we move into an era where GPU resources are scarce, and people attempt to navigate the end of Moore's law for GPU chip processing power. In short, what type of GPU(s) and how many were used for the data processing in this study.

4- Why was filtering of maximum resolution of the particles to $< 4 \text{ \AA}$ done so late in the data processing? This could have been done directly following the Patch CTF?

5- Did the authors perform Model Angelo map fitting with or without the .fasta sequence?

6- Authors should show more 2D classes and at least one step in the particle cleaning via 2D class process. Sorting particles via 3D classification can also be a way to obtain more good particles and should be considered by the authors.

7- At one point in the results it is stated that five groups who have uploaded structures or submitted bio-archive manuscripts during preparation of this manuscript, however, then later in the text this number is four... so which is it?

8- Discussion section, 2nd paragraph, 2nd sentence. The phrase 'non-trivial' in this sentence is awkward, significant differences would be more appropriate.

9- Manufacturer details for the Glacios in the materials and methods are missing

10- It is impossible to assess the degree to which the model fits the map from the data provided in the manuscript. It wasn't until the map and structure files were provided to this reviewer that this could be assessed. This is a very important point which cannot be overlooked.

11- Figure 3, it should be more obvious where your structure is in comparison. It is very difficult to tell the differences between your structure and the other structures. Using a different color scheme for the two structures would be very helpful to visualize the differences.

12- The discussion does not discuss your results and is confusing.

13- More details on how FoldX was performed are required.

14- Several side chains are extremely poor fits to the map. Even with the poor quality of this map, these fits are unreasonable and closer attention needs to be made. The same is true for the Ramachandra.

Reviewer #3 (Comments to the Authors (Required)):

Navratna et al. report the structure of the human systemic RNAi defective transmembrane protein (hSIDT1) and describe mechanistic aspects revealed by flexible regions in the structure. To this end, they compare the newly obtained structure with other previously published hSIDT1/hSIDT2 structures and another homolog from *C. elegans*.

My main criticism is that the structure reported in the manuscript seemingly shows the same conformation as the previously reported human structures. Thus, it does not provide much more insight into the mechanistic aspects of the transporter. I believe that additional comparisons with previously published structures, and examining additional features of the obtained structure, can help strengthen the study. More specifically, by addressing the following points:

1. The transmembrane domain includes the disordered cytoplasmic loop CL1 with more than 100 amino acids. The authors note that it is believed to form a cytosolic domain with an RNA binding ability. A reference for this claim is missing. In addition, is such a loop considered a typical feature for RNA binding domains in proteins?

2. The authors use the term "evolutionary conserved" multiple times throughout the manuscript. However, they do not explain how this was determined. To this end, they can use various methods, e.g., conduct multiple sequence alignments, refer to other papers, etc. A figure of the model colored by conservation score (using the ConSurf tool, for example) will allow the authors to consider the conservation of specific regions in the structure, identify the specific amino acids that are functionally conserved, and correlate them with the other findings. It will, in turn, provide further support for the mechanistic details outlined in the manuscript, and, in general, distinguish it from similar previous publications.

3. The results and the discussion sections will benefit from breaking long sections into smaller paragraphs focusing on certain

mechanistic aspects. For example, "comparing the hSIDT1 structure with cholesterol binding proteins" can include the part of the text discussing structural elements such as phenylalanine highway. "hSIDT1 is a member of the CREST superfamily" for the discussion part related to the metal ion coordination, despite its absence from the structure.

4. Are there any differences between the inter- and intra-chain interactions identified in the manuscript (Figure S4), as opposed to the ones identified in previous hSIDT1/hSIDT2 structures?
5. It is hard to understand the flexibility of different domains from Figure 3 in its current state. The difference between 2, 4, and 6Å are hard to interpret (especially in print), and makes the discussion hard to follow. The figure should include at least one example of the structure overlaid on another structure, to visually inspect the changes to the flexible parts (e.g., does it include domain-domain motion, or is it limited to more local changes? Are the changes only occurring in the flexible regions, or are there any hinge movements in the binding regions of the receptor? Etc.), and include a separate panel for the LBD superimposition.
6. The figures in the paper are not entirely clear and easy to interpret. Specifically, Figure 1 could be divided into two or even three larger figures, with better resolution, making it easier to understand. For instance, the text in panel F is difficult to read.
7. It is unclear how the superpositions were conducted (and it is not specified in the methods section). Were they all carried out by the ChimeraX tool? In addition, the sequence identity between the hSIDT1 and SID1 is not mentioned and could be useful to understand the structural variations.

Minor suggestions:

- a. It seems that the reference to Figure 4 comes before the reference to Figure 3, so switching their order can help fix this issue.
- b. I could not find the three sub-interfaces SI1, SI2, and SI3, in Figure 1. In Figure 1C, the entire DI is marked but not divided into sub-interfaces.
- c. 1 legend, in Panels (A), (B), and (C): "The LBD density is highlighted as a gray mesh in panel (c)" - should be (C).
- d. Figure 3 legend: I assume you used sequence-based pairwise alignment to superimpose the hSIDT1/2 structures, and not structure-based pairwise alignment as the text currently states.

Appeal Request

Dear Editor,

Thank you for your email. We thank the reviewers for their comments and wanted to address a few things via this email before we start working on the manuscript revisions. After carefully assessing the reviews from the referees, we wanted to explore the possibility of the following future course of action.

We believe reviewers 1 and 3 have raised valid major comments, which we can answer right away. For example, Reviewer 1 suggested local masking and variability analysis (3DVA), which we have already performed (but did not include in the manuscript). They also ask for structural comparison with PDB entries 8KCW and 8KCX. Both of these PDBs, as of today are still unreleased entries. As a result, we do not have access to these PDBs. Similarly, reviewer 3 requests clarification of the existing text and suggests editing the figures to be more legible, which we can do. In addition, we can address the minor concerns raised by all the reviewers.

However, we are perplexed by comments from reviewer 2. They seemed to have been confused about various aspects of the manuscript, such as the protein name (reviewer seems to think it is SIRT1), the microscope used (reviewer seems to think it is Glacios), and certain comments pertaining to the processing of the data, details of computational resources used, and making the PDB & map available for review along with initial submission seemed like a non-standard practice of reporting of single-particle cryo-EM structural analysis reports. Reviewer 2 also asks for smFRET analysis or Molecular Dynamics simulations and structures at higher-resolution or with cholesterol, experiments that require significant time and resources. There were six reports of SIDT family structures while we were preparing this manuscript, with two coming online while in review. Two other unreleased structures (8KCW and 8KCX) from a different study will be released anytime soon; it seems unreasonable to require us to establish new methodologies and determine the structure with different ligands. Any delay beyond a few weeks will ultimately make our work obsolete. Therefore, at this moment, we can only address issues that relate to the reanalysis of existing data and changes to the text and figures.

Please let us know if you would consider resubmitting our manuscript with the aforementioned changes. We will submit the revisions soon. We thank you again for your time and patience in handling our manuscript.

Kind regards,

Shyamal

Decision Letter for Appeal

March 21, 2024

MS: LSA-2024-02624-T

Dr. Shyamal Mosalaganti
University of Michigan
Life Sciences Institute
Mary Sue Coleman Hall
210 Washtenaw Avenue
Ann Arbor, MI 48109

Dear Dr. Mosalaganti,

Your manuscript entitled "Structure of the human SIDT1 reveals the conformational flexibility of its lipid binding domain" has now been reconsidered.

Please use the following link to submit your manuscript and rebuttal when they are ready:

<https://lsa.msubmit.net/cgi-bin/main.plex?el=A1Na3BLL7A6Crir3I3B9ftdt7vh8YAHNSL4cWIFzT7qaQZ>

Yours sincerely,

Eric Sawey, PhD
Executive Editor
Life Science Alliance

<http://www.lsjournal.org>eady present) this doesn't appear the case, the authors should clarify

Response to reviewers

We sincerely thank all three reviewers for taking the time to review our manuscript. We have incorporated as many of their suggestions as possible in the revised version of this manuscript, especially focusing on reanalyzing our data. We have made considerable changes to the text and figures per their suggestions. We agree with the general consensus amongst the reviewers regarding the discussion and results section. Hence, we rewrote significant portions of the results and discussion and added more figures. We have listed the changes in this rebuttal and provided an explanation where required. The suggestions from the reviewers have made the revised draft a much better read. Thank you!

Reviewer #1 (Comments to the Authors (Required)):

Summary

Navratna, Kumar et al. report a cryo-EM structure of the detergent-solubilized human SIDT1 in a cholesterol-free environment. In *C. elegans*, SID1 is crucial for the uptake of short non-coding RNA from the environment, leading to systemic RNA interference. These proteins are conserved in humans, despite the process of systemic RNAi not being conserved. The authors meticulously describe the general architecture of hSIDT1 and the interfaces between the two dimers and the subdomains of hSIDT1. Considering the solely structural nature of this paper and lack of any functional characterization, the submitted manuscript has limited novelty. Multiple structures of hSIDT1 and hSIDT2 were published within the last 9 months by other groups. The authors take references to these published structures by comparing their apo-structure with the previously deposited structures of hSIDT1, hSIDT2 and *C. elegans* SID1. They identify the lipid-binding domain of hSIDT1 to be flexible in the absence of cholesterol and hypothesise a potential role in lipase activity. Overall, the manuscript is well-written and the figures are accurate, but sometimes too dense with labels.

Major concerns

- The lipid binding domain of cholesterol-free hSIDT1 was described as highly flexible. The authors did not attempt to resolve the flexibility of the lipid-binding domain (LBD) of the transmembrane part of the protein. A global mask was applied during refinement, but did the authors try to refine the structure using a local mask on the LBD? Furthermore, the number of particles should allow for further classification of the particles, using either 3D classification, 3D variability analysis or 3D flex analysis in cryoSPARC or equivalent methods in other processing softwares, in order to better resolve the flexibility of the LBD.

Ans: We have included a figure below (rebuttal figure 1) showing the alternative strategies we tried to improve the resolution of the lipid-binding domain. Starting with our cleaned 302,224 particle stack (Fig. S2), we performed masked 3D classification (blue box), cryoDRGN analysis (yellow box), and Blush regularization and Dynamight

flexibility analysis (pink box). Using our best stack of 122,683 particles (Fig. S2), we also performed local refinements using a mask for only TMD (gray box). The full mask refinement yielded a 3.4 Å map (which was deposited with 8V38). The refinement with a local mask for TMD alone resulted in a relatively poor 3.9 Å structure. This figure has not been included in the main draft as none of these methods improved the resolution or resolved the LBD region of the protein better than the map which we have already deposited in the EMDB and discussed in the manuscript.

Rebuttal figure 1: (A) Masked 3D classification by applying a global mask and a local mask around TMD region. We noticed that 3D classification by applying a local TMD mask resulted in significantly poor maps ($>5 \text{ \AA}$) compared to 3D classification with a mask around the entire protein ($<5 \text{ \AA}$). The best map (with 107,490 particles) we obtained by masked 3D classification was as good ($\sim 3.4 \text{ \AA}$) as the map we deposited in EMDB. (B) cryoDRGN analysis was performed using cryoDRGN v2.3.0. The particles after homogenous refinement (C1) were imported into cryoDRGN, and the neural network was trained using a 10-dimensional latent variable and 5 layers of 1024x3 encoder and decoder architecture for 100 epochs. UMAP generated from cryoDRGN was clustered by the Gaussian-mixture model, and particles from each cluster were imported into cryoSPARC for voxel-based homogenous reconstruction followed by non-uniform and local refinement. Cluster 6 from cryoDRGN was not imported into cryoSPARC for further analysis because of the low particle number ($\sim 5,800$). The best reconstruction we obtained from cryoDRGN processing was by pooling particles from clusters 1, 4, & 5, which resulted in a 3.6 \AA map. (C) Blush regularization and DynaMight flexibility analysis. Cleaned particles were exported to Relion and were re-extracted with larger box size of 400 px. The extracted particle stack was 3D autorefined, polished, & 3D refined for Blush regularization. Flexibility within the resultant map was analyzed using DynaMight. The final map obtained by this processing strategy in Relion was at a resolution of 3.9 \AA . (D) Masked local refinement of the final subset of 122,683 particles. The structure that has been deposited was obtained by applying a mask around full protein during local refinement. When we apply a mask around the TMD region alone, the resolution is poor (3.9 \AA). A phenomenon we also observed in masked 3D classification. None of these strategies above resolved the LBD better than what is already deposited.

- The authors state that the observed flexible LBD in the cholesterol-free structure facilitates the lipase activity and base this on the structure comparison with other hSIDT1/hSIDT2 structures. However, the comparison of the structures leads to the conclusion that the LBD of apo hSIDT1 is nearly identical to the LBD of the mutant E555Q, a mutant with abrogated lipase activity. This conclusion seems contradictory, as the facilitation of lipase activity and the abrogated lipase activity are contrary.

Ans: We agree with the reviewer on this comment. While preparing to resubmit this draft, another study reporting the structures of hSIDT1 at two pHs became available (8WOT at pH 5.5 and 8WOR at pH 7.5). After including these structures in our structure comparison analysis, we noticed that LBD of our structure (8V38) superposes well with 8JUN (lipase inactive E555Q-hSIDT1 at pH 7.5) and 8WOT (hSIDT1-cholesterol complex at pH 5.5). At the same time, we notice that hSIDT1-phosphatidic acid complex at pH 7.5 (8JUL) superposes relatively well with a hSIDT1-sphingolipid-cholesterol structure obtained at pH 7.5 (8WOR). And, both 8JUL and 8WOR LBDs show significant differences upon superposition with 8V38. Except for our study, all the other hSIDT1 structures that are included in the comparison have been purified in the presence of CHS (Table S1). Based on our reanalysis, we conclude that LBD dynamics could not be correlated to either pH or cholesterol binding alone. While cholesterol may stabilize the TMD, our re-analysis revealed that LBD motion occurs upon phospholipid binding at the LBD and not because of pH or cholesterol. We have rewritten the results and discussion as per the reviewer's suggestions to reflect these changes and modified our conclusion. We also edited structure superposition figures (Fig. 5, Fig. 6, and Fig. S5) discussing these findings.

- The structure comparison is incomplete. There are two more hSIDT1 structures available on PDB (8KCW and 8KCX), which were not included in the comparison. Could the authors include these two comparisons into Figure 3 and the main text?

Ans: Both 8KCW and 8KCX structures discussed by Hirano et al. have been cited in the manuscript & Table S1. However, when composing this rebuttal, they are still marked as unreleased PDB entries. Thus, we could not include their comparison in the revised manuscript. Instead, we included a comparison with two new SIDT1 PDBs – 8WOT and 8WOR reported by Liu et al which became available while we revised our manuscript.

Minor concerns

- Title for Figure 2: "Critical inter-chain and intra-chain interactions of hSIDT1" - There is no biochemical evidence for the interactions to be critical. Perhaps a more modest title would be a better fit.

Ans: The title has been changed in the revised version as per the reviewer's suggestion.

- The panels in Figure 2A and 2D are hard to read due to the many labels.

Ans: We have removed all the redundant labels and changed the positions of existing ones to make the figures more legible. Fig. 2 is now Fig. 3 in the revised manuscript.

- Figure S1 - panel F - Could the authors comment on the second band that is visible on the gel?

Ans: The two bands that we see in panel F are differently glycosylated forms of hSIDT1-GFP. We have performed peptide mass fingerprinting analysis on both the excised bands from the gel and confirmed that they are both hSIDT1-GFP.

- Chapter 2.3. Comparison of hSIDT1 structure with its homologs - the authors write "we used TMD core of these proteins to superpose chain A of each structure onto our hSIDT1 map (RCSB PDB: 8V38)" but they are not referring to a map rather to the corresponding model.

Ans: Yes, that is an error, which have corrected in the revised version.

- "A noteworthy observation is the position of COMs of the ECD and full chains of the E555Q hSIDT1 mutant (8JUN) in comparison with wild-type hSIDT1 (Fig. 5B and 6C)." - There is no Figure 6C.

Ans: We have corrected this typo in the revised manuscript.

- Discussion - the authors state the differences in the purification and vitrification conditions between their work and the other published work, however they do not explain satisfyingly what may cause the differences.

Ans: In the revised version of the manuscript, we have added a few lines to this paragraph highlighting the differences in our sample prep compared to others, and how that could have led to a poorly resolved LBD. We believe, the primary reason for a poorly resolved LBD may be the lack of cholesterol or CHS in our prep.

- Table 1 - Validation statistics - the model contains 0.6% Ramachandran Outliers. This needs to be improved to reduce the statistics to 0% Ramachandran outliers, unless the outliers are supported by high resolution density.

Ans: We agree with reviewer's concern here. We have deleted more side chains, and loops within the LBD, with bad density, and modeled our structure to reduce Ramachandran outliers.

Referee Cross-Comments

I agree with the other referees comments.

Reviewer #2 (Comments to the Authors (Required)):

Comments to the Authors:

While I am sure it is disappointing to have other publications emerge as you are working on analysis and publication of your hSIRT1 data, in its current state, this manuscript does not add significant findings to the field. I have made this decision on the grounds that the paper contains only structural data, and that data is not of a high enough quality to support the conclusions made. While I appreciate that the authors have attempted to add novelty by comparing their structure with other recently available structures and performing predictive mutagenesis on the SIRT1 protein, the cryo-EM data produced by the authors was not of high enough quality for these comparisons to be reliable or meaningful.

A short summary of the paper and main points, including description of the advance offered to the field.

This manuscript provides a Cryo-EM map of hSIRT1 protein at 3.4 Å (3.98 Å unmasked), due to the resolution of the map the authors fit a model and determine a structure of the protein hSIRT1. They also compare their SIRT1 structure to others in the field and model some mutations in the protein.

Ans: The structure reported and discussed in this manuscript is of hSIDT1 and not hSIRT1.

The main novel findings the authors are claiming is:

1) That the lipid binding domain was unable to be resolved because they isolated the

protein without cholesterol, and this made the lipid binding domain too flexible to resolve.

It is stated that "TMs 5-9 form a conformationally flexible domain which is poorly resolved in our cryo-EM map" in your LBD results section. This is a vast understatement. These regions aren't poorly resolved, they are actually absent for the most part. It is not possible to model this region of the protein with the quality of the map and given the major conclusions of the paper surround this region, these conclusions cannot be substantiated from these data.

Ans: We see poor local resolution for cytosolic halves of TM5 and TM9, and hence we modeled only main chain for these regions of the LBD based on the sharpened map that is deposited in EMDB as the primary map. For the TMs 6-8, we only see poor discontinuous density in the sharpened map. However, in the unsharpened map, which is also deposited in the EMDB as additional map, we see density to be able to model main chains of these TMs. We have made a note of these caveats in the section 2.1 of the results, and the legends of Fig. S2 and Fig. S3.

While the hypothesis the authors make regarding the flexibility of the LBD is reasonable to make, for this claim to be substantiated, the following (significant) steps would need to be addressed:

1- Re-analyze the cryo-EM data/ collect additional data and perform further analysis e.g. 3DVA/3D flex.

Ans: We have included a figure (see rebuttal figure 1) showing the alternative strategies we tried to improve resolution. Starting with our cleaned 302,224 particle stack (Fig. S2), we performed masked 3D classification (blue box), cryoDRGN analysis (yellow box), and Blush regularization and Dynamight flexibility analysis (pink box). Using our best stack of 122,683 particles (Fig. S2), we also performed local refinements using a mask for only TMD (gray box). The full mask refinement yielded a 3.4 Å map (which was deposited with 8V38). The refinement with a local mask for TMD alone resulted in a relatively poor 3.9 Å structure. This figure has not been included in the main draft as none of these methods improved the resolution or resolved the LBD region of the protein better than the map which we have already deposited in the EMDB and discussed in the manuscript.

Rebuttal figure 1: (A) Masked 3D classification by applying a global mask and a local mask around TMD region. We noticed that 3D classification by applying a local TMD mask resulted in significantly poor maps ($>5\text{ \AA}$) compared to 3D classification with a mask around the entire protein ($<5\text{ \AA}$). The best map (with 107,490 particles) we obtained by masked 3D classification was as good ($\sim 3.4\text{ \AA}$) as the map we deposited in EMDB. (B) cryoDRGN analysis was performed using cryoDRGN v2.3.0. The particles after homogenous refinement (C1) were imported into cryoDRGN, and the neural network was trained using a 10-dimensional latent variable and 5 layers of 1024×3 encoder and decoder architecture for 100 epochs. UMAP

generated from cryoDRGN was clustered by the Gaussian-mixture model, and particles from each cluster were imported into cryoSPARC for voxel-based homogenous reconstruction followed by non-uniform and local refinement. Cluster 6 from cryoDRGN was not imported into cryoSPARC for further analysis because of the low particle number (~5,800). The best reconstruction we obtained from cryoDRGN processing was by pooling particles from clusters 1, 4, & 5, which resulted in a 3.6 Å map. **(C)** Blush regularization and DynaMight flexibility analysis. Cleaned particles were exported to Relion and were re-extracted with larger box size of 400 px. The extracted particle stack was 3D autorefined, polished, & 3D refined for Blush regularization. Flexibility within the resultant map was analyzed using DynaMight. The final map obtained by this processing strategy in Relion was at a resolution of 3.9 Å. **(D)** Masked local refinement of the final subset of 122,683 particles. The structure that has been deposited was obtained by applying a mask around full protein during local refinement. When we apply a mask around the TMD region alone, the resolution is poor (3.9 Å). A phenomenon we also observed in masked 3D classification. None of these strategies above resolved the LBD better than what is already deposited.

2- Collect data with cholesterol bound hSIRT1 to demonstrate that the lack of cholesterol here is specifically the cause of the low resolution (rather than the quality of the cryo-EM data or stability of the membrane protein in the detergent membrane mimetic of choice).

Ans: All hSIRT1 structures that we compared in this report have been obtained in detergent micelles but in the presence of cholesterol or CHS, except our structure. All other structures show well-resolved LBD. Based on this structure comparison, we propose that cholesterol binding potentially stabilizes the TMD. However, our structure comparison reanalysis suggests that binding of cholesterol does not cause conformational change in LBD, but the movement in LBD is primarily induced by phospholipid binding at the lipase site. See Fig. 5, Fig. 6, and Fig. S7.

3- If you want to test how this domain moves, you need a better structure and to run molecular dynamics simulations.

4- Perform additional experiments to analyze movement of this domain in the presence or absence of cholesterol (e.g. smFRET/molecular dynamics).

2) That some modelled mutations affect the structure of the protein. These mutations should be made, and functionally assessed.

Ans: We agree with all the issues raised by this reviewer in points 3 & 4 above. We are conducting more experimental exploration via structural characterization of ligand bound hSIRT1, molecular dynamics, and mutational analysis, to pinpoint the molecular basis of conformational dynamics in hSIRT1. However, we consider these efforts to be outside the scope of this current manuscript. As a result, we rewrote sections of discussion based on the results that we included in the manuscript.

The final statement in the discussion is a major issue:

"we hypothesize that the RNA binding at the JR1-JR2 interface induces a rearrangement of JR1 and JR2 which allows for the formation of an opening that can

feed RNA to the TMD. Then, the metal ion dependent lipase activity in the TMD region disturbs the lipid bilayer around the protein and allows for the LBD and TM2 to rearrange, enabling the formation of a cavity large enough to allow the passage of RNA to cytosol."

Again, like the issues of the results, the authors cannot make this prediction with the cryo-EM data at the level of quality presented in this paper or without significant additional experiments (as outlined above) to warrant proposing this hypothesis.

Ans: We agree with the reviewer, and based on the suggestions from other reviewers, we rewrote a new conclusion statement in the discussion. We have also compared our structure with two more hSIDT1 structures and provided a rationale for changes in the conformation of LBD.

Additional issues to be addressed

1- The second paragraph of the introduction, 3rd sentence is missing an 'a' it should be "that is a popular cancer therapeutic"

Ans: We have rephrased this sentence in the revised version.

2- Watch the consistency of you naming of HEK293S GnTi-, pick a name and stick with it

Ans: We have consistently used 'HEK293S GnTi-' in all our methods & results. The only other instance of HEK cell usage in text appears in discussion, when citing Qian et al., study, where they used HEK293F cells for their expression.

3- The specifications of the machine or server the data was processed on. While unconventional, open sharing of this information should become the gold standard as we move into an era where GPU resources are scarce, and people attempt to navigate the end of Moore's law for GPU chip processing power. In short, what type of GPU(s) and how many were used for the data processing in this study.

Ans: We have not included this information in the methods section, because it appeared to be both unconventional and uninformative, as the computation power used varies with respect to the job or software being run. The data was processed on the University of Michigan Life Sciences Institute cluster which has 4 each of NVIDIA RTX 2080 Ti (11 GB memory/GPU) and NVIDIA A40 (48 GB memory/GPU) GPUs and has a storage space of about 4TB.

4- Why was filtering of maximum resolution of the particles to $< 4 \text{ \AA}$ done so late in the data processing? This could have been done directly following the Patch CTF?

Ans: We did not anticipate that curating cleaned particles from micrographs with a CTF fit of $< 4 \text{ \AA}$ would yield a better map, right at the beginning of the processing. In addition,

it has been demonstrated by several groups that it is possible to obtain a high-resolution map without having to curate the particle stack using the <4 Å CTF fit criteria.

5- Did the authors perform Model Angelo map fitting with or without the .fasta sequence?

Ans: ModelAngelo was used to build hSIDT1 into the map using fasta sequence as input. However, only the ECD was built accurately in the resultant output. We replaced the TMD core of this with AlphaFold hSIDT1 model and built parts of LBD manually in Coot.

6- Authors should show more 2D classes and at least one step in the particle cleaning via 2D class process. Sorting particles via 3D classification can also be a way to obtain more good particles and should be considered by the authors.

Ans: Including more 2D classification steps was not adding more information to the processing figure presented in Fig. S2. We have employed iterative 2D classification and heterogenous refinements to identify the cleanest particle set. In the rebuttal, we have included a figure discussing alternative masked 3D classification attempts that we had made at improving the particle stack quality.

7- At one point in the results it is stated that five groups who have uploaded structures or submitted bio-archive manuscripts during preparation of this manuscript, however, then later in the text this number is four... so which is it?

Ans: The C. elegans homologs have been deposited in RCSB but they do not have a preprint, or a published manuscript, associated with the deposition. As a result, in Table S1 we compared the methods section of only the published reports, and for the figures we used only deposited and released structures. To avoid further confusion, we rephrased the text to read "other groups".

8- Discussion section, 2nd paragraph, 2nd sentence. The phrase 'non-trivial' in this sentence is awkward, significant differences would be more appropriate.

Ans: We have edited out the phrase "non-trivial".

9- Manufacturer details for the Glacios in the materials and methods are missing

Ans: The data was collected on Titan Krios G4i cryo-TEM, and details of data collection are mentioned in the methods section.

10- It is impossible to assess the degree to which the model fits the map from the data provided in the manuscript. It wasn't until the map and structure files were provided to this reviewer that this could be assessed. This is a very important point which cannot be overlooked.

Ans: There was no mechanism to deposit a large file like the EM map along with the initial submission. Thus, we provided the validation report with initial submission, and followed the standard operating procedure of submitting the map and coordinates upon request.

11- Figure 3, it should be more obvious where your structure is in comparison. It is very difficult to tell the differences between your structure and the other structures. Using a different color scheme for the two structures would be very helpful to visualize the differences.

Ans: Fig. 3 (Fig. 5 and Fig. S7 in the revised version) does not show structure superpositions directly. As mentioned in the results and legends, the panels in structure superposition figure show chain A of hSIDT1 structure colored based on the $C\alpha$ RMSD obtained upon aligning it with various structures listed in Table S2. However, both the figure and legend have been now edited to make this clear and we have also included the structure being compared in gray in all the panels of Fig. 5 and Fig. S7.

12- The discussion does not discuss your results and is confusing.

Ans: We have rephrased the results and discussion in the revised version of the manuscript for better clarity.

13- More details on how FoldX was performed are required.

Ans: In the revised version of the manuscript, we removed the figure & discussion relating to our FoldX analysis, as we do not have biochemical or structure-based evidence to corroborate FoldX predictions.

14- Several side chains are extremely poor fits to the map. Even with the poor quality of this map, these fits are unreasonable and closer attention needs to be made. The same is true for the Ramachandra.

Ans: We agree with reviewer's concern here. We have deleted more side chains, and loops within the LBD, with bad density, and modeled our structure to reduce Ramachandran outliers to 0%.

Reviewer #3 (Comments to the Authors (Required)):

Navratna et al. report the structure of the human systemic RNAi defective transmembrane protein (hSIDT1) and describe mechanistic aspects revealed by flexible regions in the structure. To this end, they compare the newly obtained structure with other previously published hSIDT1/hSIDT2 structures and another homolog from *C. elegans*.

My main criticism is that the structure reported in the manuscript seemingly shows the same conformation as the previously reported human structures. Thus, it does not

provide much more insight into the mechanistic aspects of the transporter. I believe that additional comparisons with previously published structures, and examining additional features of the obtained structure, can help strengthen the study. More specifically, by addressing the following points:

1. The transmembrane domain includes the disordered cytoplasmic loop CL1 with more than 100 amino acids. The authors note that it is believed to form a cytosolic domain with an RNA binding ability. A reference for this claim is missing. In addition, is such a loop considered a typical feature for RNA binding domains in proteins?

Ans: We had included the references for this in the discussion. But, we have now added the references to this part of the results section in the revised version of the manuscript. While cytoplasmic loop of SIDT family of proteins is not considered a canonical RNA binding domain, Hase, K. et al. (Autophagy, 2020) demonstrate that SIDT2 cytosolic loop has a nucleic acid binding Arginine Rich Motif (ARM) similar to ARMs found in other RNA binding proteins, such as LAMP2C.

2. The authors use the term "evolutionary conserved" multiple times throughout the manuscript. However, they do not explain how this was determined. To this end, they can use various methods, e.g., conduct multiple sequence alignments, refer to other papers, etc. A figure of the model colored by conservation score (using the ConSurf tool, for example) will allow the authors to consider the conservation of specific regions in the structure, identify the specific amino acids that are functionally conserved, and correlate them with the other findings. It will, in turn, provide further support for the mechanistic details outlined in the manuscript, and, in general, distinguish it from similar previous publications.

Ans: We have included a new supplementary figure (Fig. S8C), in the revised version of the manuscript, depicting AlphaFold model of hSIDT1 colored as per the evolutionary sequence conservation score obtained from ConSurf server. We have also included a superposition of AlphaFold models of hSIDT1 and hSIDT1 cryo-EM structure discussed in this report. We have included more structure comparison and conserved features in the results and discussion sections of the revised version of the manuscript, including more figures (Fig. 5C, Fig. 5D, Fig. 6D, and Fig. S6).

3. The results and the discussion sections will benefit from breaking long sections into smaller paragraphs focusing on certain mechanistic aspects. For example, "comparing the hSIDT1 structure with cholesterol binding proteins" can include the part of the text discussing structural elements such as phenylalanine highway. "hSIDT1 is a member of the CREST superfamily" for the discussion part related to the metal ion coordination, despite its absence from the structure.

Ans: We agree with the reviewer's comment. We have rewritten the results and discussion for brevity.

4. Are there any differences between the inter- and intra-chain interactions identified in

the manuscript (Figure S4), as opposed to the ones identified in previous hSIDT1/hSIDT2 structures?

Ans: In our structure comparison analysis, we observed that the intra-chain interactions are conserved between SIDT1 and SIDT2. Regarding inter-chain interactions, the side chains involved in hydrogen bonding interactions at the ECD-A and ECD-B interface are conserved. While the residues themselves are conserved at the TMD-A and TMD-B interface, the hydrogen bonding network or stacking interactions vary because of the conformationally dynamic TMDs. We have included new figures and text in the revised draft of the manuscript to show that these sites lie in the evolutionarily conserved regions of the protein and the side chains that contribute to the inter- and intra-chain interactions are highlighted (Fig. 6, Fig. S5, Fig. S6, Fig. S8).

5. It is hard to understand the flexibility of different domains from Figure 3 in its current state. The difference between 2, 4, and 6Å are hard to interpret (especially in print), and makes the discussion hard to follow. The figure should include at least one example of the structure overlaid on another structure, to visually inspect the changes to the flexible parts (e.g., does it include domain-domain motion, or is it limited to more local changes? Are the changes only occurring in the flexible regions, or are there any hinge movements in the binding regions of the receptor? Etc.), and include a separate panel for the LBD superimposition.

Ans: We agree with the reviewers' critic here. We have now edited this figure and included it in two parts, Fig. 5 and Fig. S7, in the new draft. We have also included the structure being superposed with our structure (8V38) in each of the panels in gray and showed LBD superposition alone in a separate panel. In addition, we added more structure comparison analyses to provide a context for movement in LBD and TMD regions of SIDT1 and SIDT2 (Fig. 6D, Fig. 6E, Fig. S5, and Fig. S6).

6. The figures in the paper are not entirely clear and easy to interpret. Specifically, Figure 1 could be divided into two or even three larger figures, with better resolution, making it easier to understand. For instance, the text in panel F is difficult to read.

Ans: We have split figure 1 into two parts as suggested (Fig. 1 and Fig. 2). In addition, we edited the labels and also included more figures in the revised version of the manuscript.

7. It is unclear how the superpositions were conducted (and it is not specified in the methods section). Were they all carried out by the ChimeraX tool? In addition, the sequence identity between the hSIDT1 and SID1 is not mentioned and could be useful to understand the structural variations.

Ans: We have elaborated more on how the superpositions were made, in results and the figure legends. The superpositions were all performed in ChimeraX, and we have added this description in the revised methods section. In the revised version of the manuscript, we have also included a figure of hSIDT1 model colored by evolutionary

sequence conservation in the supplementary section (Fig. S8C) to reflect the sequence identity within hSIDT1 homologs and included a figure comparing conserved interactions of hSIDT1 and hSIDT2 (Fig. S6).

Minor suggestions:

a. It seems that the reference to Figure 4 comes before the reference to Figure 3, so switching their order can help fix this issue.

Ans: We have switched the order of these figures to reflect the order of their appearance in the text.

b. I could not find the three sub-interfaces SI1, SI2, and SI3, in Figure 1. In Figure 1C, the entire DI is marked but not divided into sub-interfaces.

Ans: To minimize the crowding of labels, we discussed the dimer interface predominantly in Figure 2 (Fig. 3 in the revised version). To minimize the confusion, we have edited the figure callout in section 2.2.3.

c. 1 legend, in Panels (A), (B), and (C): "The LBD density is highlighted as a gray mesh in panel (c)" - should be (C).

Ans: We have corrected this typo in the revised version of the manuscript.

d. Figure 3 legend: I assume you used sequence-based pairwise alignment to superimpose the hSIDT1/2 structures, and not structure-based pairwise alignment as the text currently states.

Ans: Yes, that is correct. We used pairwise sequence alignment tool in ChimeraX, and not structure-based pairwise alignment as mentioned. The legend & methods section have been now changed to reflect this.

May 31, 2024

RE: Life Science Alliance Manuscript #LSA-2024-02624-TR-A

Dr. Shyamal Mosalaganti
University of Michigan-Ann Arbor
Life Sciences Institute
Mary Sue Coleman Hall
210 Washtenaw Avenue
Ann Arbor, MI 48109

Dear Dr. Mosalaganti,

Thank you for submitting your revised manuscript entitled "Structure of the human SIDT1 reveals the conformational flexibility of its lipid binding domain". We would be happy to publish your paper in Life Science Alliance pending final revisions necessary to meet our formatting guidelines.

- please address the Reviewers' remaining comments
- please be sure that the authorship listing and order is correct
- please upload your main manuscript text as an editable doc file
- please upload all figure files as individual ones, including the supplementary figure files; all figure legends should only appear in the main manuscript file
- please add a Running Title and a Summary Blurb/Alternate Abstract to our system
- please add ORCID ID for the secondary corresponding author -- they should have received instructions on how to do so
- please add Keywords and Categories for your manuscript to our system
- please add the Twitter handle of your host institute/organization as well as your own or/and one of the authors in our system
- please be sure to add all authors to the author contributions section of your paper
- please add your main, supplementary figure, and table legends to the main manuscript text after the references section
- please upload your Tables in editable .doc or excel format. They can be included at the bottom of the main manuscript file or sent as separate files.
- please incorporate supplementary references into the main references section in the manuscript file
- please add callouts for Figures 2B; S5E and S7A-C to your main manuscript text

A. FINAL FILES:

-- Summary blurb (enter in submission system): A short text summarizing in a single sentence the study (max. 200 characters including spaces). This text is used in conjunction with the titles of papers, hence should be informative and complementary to the title. It should describe the context and significance of the findings for a general readership; it should be written in the

present tense and refer to the work in the third person. Author names should not be mentioned.

B. MANUSCRIPT ORGANIZATION AND FORMATTING:

Sincerely,

Reviewer #1 (Comments to the Authors (Required)):

Review -

Structure of the human systemic RNAi defective transmembrane protein 1 (hSIDT1) reveals the conformational flexibility of its lipid binding domain

- Navratna, Kumar et al. 2024

Summary

Navratna, Kumar et al. resubmit their report on a cryo-EM structure of the detergent-solubilized human SIDT1 in a cholesterol-free environment. In *C. elegans*, SID1 is crucial for the uptake of short non-coding RNA from the environment, leading to systemic RNA interference. These proteins are conserved in humans, despite the process of systemic RNAi not being conserved. The authors meticulously describe the general architecture of hSIDT1 and the interfaces between the two dimers and the subdomains of hSIDT1. Multiple structures of hSIDT1 and hSIDT2 were published within the last 9 months by other groups. The authors take references to these published structures by comparing their apo-structure with the previously deposited structures of hSIDT1, hSIDT2 and *C. elegans* SID1. They identify the lipid-binding domain of hSIDT1 to be the most flexible part of the protein. The findings are phrased in a more conservative way compared to the first submission, which fits to the solely structural nature of the paper.

Major concerns

No major concerns raised in the revised version of the manuscript.

Minor concerns

- Figure 2A and 3: boxes highlighting different regions of the protein should either be a bit thicker or different colors. It is difficult to distinguish the colors currently.

- Figure 6 and Figure S5: Difficult to distinguish the different shades of pink and yellow. A different choice of colors might be beneficial.

- The explanations on the observed differences due to the differences in sample prep (now emphasizing on the pH and the higher temperature before plunging) are still not satisfying.

Reviewer #3 (Comments to the Authors (Required)):

Mosalaganti and co-workers present a revised draft of LSA-2024-02624-T. All three reviewers noted the lack of significant findings that would justify the publication of the manuscript in its original form and provided various suggestions to address this issue. I found the revised manuscript, and especially the discussion section, written in a much clearer way than the original submission. In addition, the figures, which previously were overcrowded and, in several instances, very difficult to follow, are now much easier to understand. It is evident that the authors put much effort in revising the manuscript following the reviewers' comments. The following suggestions will help highlight and better communicate the main findings of the work:

- Previously published structures of hSIDT1 proteins used cholesteryl hemisuccinate (CHS) during solubilization, unlike the new structure derived by the authors. In the discussion, the authors suggest that this could contribute to the differences observed between their structure (RCSB PDB 8V38) and previous publications, and most importantly, could contribute to their ability to get satisfying resolution for the lipid binding domain (LBD). Moreover, they claim that the LBD is the most dynamic region of the protein, and that the presence of a cholesterol is the main influence on the conformational state of the LBD - which is not surprising.
 - o Since the authors mention varying degree of motion of the LBD domain, I would expect Figure 5 to include an all-vs-all structure alignment of this region, or at least to show the largest changes between two structures in terms of RMSD.
 - o Also, the authors should include a discussion of what could be the cause of such motion. Even if further investigation is required, what changes to the current methodology are needed to allow for this further investigation?
- It is not clear to me why the authors chose the AlphaFold model for displaying the evolutionary sequence conservation (Figure S8C), instead of using the actual structure derived by the authors or one of the other published structures.
- In Figure 2, the blue and black boxes colors are hard to tell apart. I suggest using a light color for one of the boxes.
- "We also identify a phenylalanine highway within the TMD core, that is often seen in ATP-binding cassette sterol transporters" - reference needed.

Response to reviewers (round 2)

Reviewer #1 (Comments to the Authors (Required)):

Review –

Structure of the human systemic RNAi defective transmembrane protein 1 (hSIDT1) reveals the conformational flexibility of its lipid binding domain
- Navratna, Kumar et al. 2024

Summary

Navratna, Kumar et al. resubmit their report on a cryo-EM structure of the detergent-solubilized human SIDT1 in a cholesterol-free environment. In *C. elegans*, SID1 is crucial for the uptake of short non-coding RNA from the environment, leading to systemic RNA interference. These proteins are conserved in humans, despite the process of systemic RNAi not being conserved. The authors meticulously describe the general architecture of hSIDT1 and the interfaces between the two dimers and the subdomains of hSIDT1. Multiple structures of hSIDT1 and hSIDT2 were published within the last 9 months by other groups. The authors take references to these published structures by comparing their apo-structure with the previously deposited structures of hSIDT1, hSIDT2 and *C. elegans* SID1. They identify the lipid-binding domain of hSIDT1 to be the most flexible part of the protein. The findings are phrased in a more conservative way compared to the first submission, which fits to the solely structural nature of the paper.

Major concerns

No major concerns raised in the revised version of the manuscript.

Minor concerns

- Figure 2A and 3: boxes highlighting different regions of the protein should either be a bit thicker or different colors. It is difficult to distinguish the colors currently.

Ans: We have edited the box thickness and colors in all the figures in the revised manuscript.

- Figure 6 and Figure S5: Difficult to distinguish the different shades of pink and yellow. A different choice of colors might be beneficial.

Ans: The choices of colors were from a colorblind friendly palette. To aid in easier differentiation, all hSIDT1 spheres in Fig. 6 have been highlighted by a black border.

- The explanations on the observed differences due to the differences in sample prep (now emphasizing on the pH and the higher temperature before plunging) are still not satisfying.

Ans: We have added another couple of lines to the discussion explaining how our sample prep condition might've contributed to differences in the structure compared to other reported SIDT1 structures.

Reviewer #3 (Comments to the Authors (Required)):

Mosalaganti and co-workers present a revised draft of LSA-2024-02624-T. All three reviewers noted the lack of significant findings that would justify the publication of the manuscript in its original form and provided various suggestions to address this issue. I found the revised manuscript, and especially the discussion section, written in a much clearer way than the original submission. In addition, the figures, which previously were overcrowded and, in several instances, very difficult to follow, are now much easier to understand. It is evident that the authors put much effort in revising the manuscript following the reviewers' comments. The following suggestions will help highlight and better communicate the main findings of the work:

- Previously published structures of hSIDT1 proteins used cholesteryl hemisuccinate (CHS) during solubilization, unlike the new structure derived by the authors. In the discussion, the authors suggest that this could contribute to the differences observed between their structure (RCSB PDB 8V38) and previous publications, and most importantly, could contribute to their ability to get satisfying resolution for the lipid binding domain (LBD). Moreover, they claim that the LBD is the most dynamic region of the protein, and that the presence of a cholesterol is the main influence on the conformational state of the LBD - which is not surprising.

o Since the authors mention varying degree of motion of the LBD domain, I would expect Figure 5 to include an all-vs-all structure alignment of this region, or at least to show the largest changes between two structures in terms of RMSD.

Ans: We have color coded chain of hSIDT1 using average RMSD of all-vs-all structure alignment and included this as a new panel in supplementary figures (Fig. S7F). The highest degree of motion in that case too is in the LBD region.

o Also, the authors should include a discussion of what could be the cause of such motion. Even if further investigation is required, what changes to the current methodology are needed to allow for this further investigation?

Ans: We have added a couple of lines of text to the discussion addressing this, in the revised version.

- It is not clear to me why the authors chose the AlphaFold model for displaying the evolutionary sequence conservation (Figure S8C), instead of using the actual structure derived by the authors or one of the other published structures.

Ans: As described in the manuscript, we were not able to build the entire model because of poor density in multiple regions. Evolutionary sequence conservation-based

coloring of our cryo-EM model by ConSurf was inefficient, thus we used AlphaFold model for representing evolutionary sequence conservation.

- In Figure 2, the blue and black boxes colors are hard to tell apart. I suggest using a light color for one of the boxes.

Ans: We changed the color of the box highlighting ECD to yellow.

- "We also identify a phenylalanine highway within the TMD core, that is often seen in ATP-binding cassette sterol transporters" - reference needed.

Ans: A reference for this has been already included in the previous versions. We have now called out that reference at this sentence's ending as well.

June 12, 2024

RE: Life Science Alliance Manuscript #LSA-2024-02624-TRR

Dr. Shyamal Mosalaganti
University of Michigan-Ann Arbor
Life Sciences Institute
Mary Sue Coleman Hall
210 Washtenaw Avenue
Ann Arbor, MI 48109

Dear Dr. Mosalaganti,

Thank you for submitting your Research Article entitled "Structure of the human SIDT1 reveals the conformational flexibility of its lipid binding domain". It is a pleasure to let you know that your manuscript is now accepted for publication in Life Science Alliance. Congratulations on this interesting work.

DISTRIBUTION OF MATERIALS:

Again, congratulations on a very nice paper. I hope you found the review process to be constructive and are pleased with how the manuscript was handled editorially. We look forward to future exciting submissions from your lab.

Sincerely,
